# PharmacoSTORM nanoscale pharmacology reveals cariprazine binding on Islands of Calleja granule cells

Susanne Prokop [1,2,11], Péter Ábrányi-Balogh [3,11], Benjámin Barti [1,2,4], Márton Vámosi [1], Miklós Zöldi [1,2,4], László Barna[5], Gabriella M. Urbán[1], András Dávid Tóth [6,7], Barna Dudok [1,8], Attila Egyed[3], Hui Deng[9], Gian Marco Leggio[10], László Hunyady[6,7], Mario van der Stelt [9], György M. Keserű [3,11] & István Katona [1,4✉]

Immunolabeling and autoradiography have traditionally been applied as the methods-of-choice to visualize and collect molecular information about physiological and pathological processes. Here, we introduce PharmacoSTORM super-resolution imaging that combines the complementary advantages of these approaches and enables cell-type- and compartment-specific nanoscale molecular measurements. We exploited rational chemical design for fluorophore-tagged high-affinity receptor ligands and an enzyme inhibitor; and demonstrated broad PharmacoSTORM applicability for three protein classes and for cariprazine, a clinically approved antipsychotic and antidepressant drug. Because the neurobiological substrate of cariprazine has remained elusive, we took advantage of PharmacoSTORM to provide in vivo evidence that cariprazine predominantly binds to $D_3$ dopamine receptors on Islands of Calleja granule cell axons but avoids dopaminergic terminals. These findings show that Pharmaco-STORM helps to quantify drug-target interaction sites at the nanoscale level in a cell-type- and subcellular context-dependent manner and within complex tissue preparations. Moreover, the results highlight the underappreciated neuropsychiatric significance of the Islands of Calleja in the ventral forebrain.

[1] Momentum Laboratory of Molecular Neurobiology, Institute of Experimental Medicine, Budapest, Hungary. [2] School of Ph.D. Studies, Semmelweis University, Budapest, Hungary. [3] Medicinal Chemistry Research Group, Research Centre for Natural Sciences, Budapest, Hungary. [4] Department of Psychological and Brain Sciences, Indiana University, Bloomington, IN, USA. [5] Nikon Center of Excellence for Neuronal Imaging, Institute of Experimental Medicine, Budapest, Hungary. [6] Department of Physiology, Faculty of Medicine, Semmelweis University, Budapest, Hungary. [7] MTA-SE Laboratory of Molecular Physiology, Eötvös Loránd Research Network, Budapest, Hungary. [8] Department of Neurosurgery, Stanford University, Stanford, CA, USA. [9] Department of Molecular Physiology, Leiden Institute of Chemistry, Leiden University & Oncode Institute, Leiden, the Netherlands. [10] Department of Biomedical and Biotechnological Sciences, Section of Pharmacology, University of Catania, Catania, Italy. [11] These authors contributed equally: Susanne Prokop, Péter Ábrányi-Balogh, György M. Keserű. ✉email: katona@koki.hu

Deciphering the molecular organization of signaling processes and their disorganization in disease remain among the greatest challenges in biomedical science. Perhaps the best example to illustrate the methodical difficulties is the molecular analysis of the brain, our most complex organ. Brain circuits consist of more than hundred distinct neuronal and glial cell types[1], which communicate via billions of synapses built up from highly compartmentalized nanoscale signaling platforms[2]. Activity-dependent changes in the density of these signalosomes occur at the nanoscale level and are fundamentally important for physiological and pathological brain plasticity[3]. Importantly, molecular alterations often swing into opposite directions in a cell- and subcellular compartment-specific manner. However, conventional ensemble approaches that are widely used in anatomy, biochemistry, and pharmacology for the quantification of molecular changes are limited in their capabilities to capture the multiscale level of immense diversity in the brain and in other organs.

The prevailing method for molecular imaging is antibody-based immunolabeling. The invention of single-molecule localization microscopy[4–6] and its application to immunolabeling in brain tissue revealed numerous details about the molecular nanoarchitecture of subcellular compartments in neuronal circuits[7–9]. However, several major limitations of immunolabeling still represent significant hurdles. To visualize new target proteins, a major drawback is usually the lack of sensitive and specific antibodies. Conformational epitopes often remain inaccessible to antibodies within crowded molecular complexes, whereas off-target interactions may lead to false-positive results in complex tissue preparations. Accordingly, systematic analysis of several thousand experimental antibodies revealed a generally poor performance necessitating laborious quality control approaches[10,11]. Moreover, quantitative analysis of immunolabeling is hindered by the batch-to-batch antibody variability, the diverging efficiency of signal amplification steps, and the uneven antibody diffusion within chemically fixed tissue samples[12]. Antibodies can label immature or inactive proteins hence their usefulness to quantify functionally relevant pools of signaling molecules is also limited. These methodical obstacles together introduce substantial technical variability and may even contribute to erroneous diagnostic interpretations in the clinical use of immunostaining[13]. Another long-standing approach to visualize and quantify molecular distribution patterns is radiolabeling-based autoradiography[14]. Small molecules readily penetrate tissue preparations and bind their targets with known stoichiometry. Therefore, autoradiography is an excellent tool for pharmacologists for the quantitative characterization of in vivo drug binding with subsequent ex vivo measurements in sections, whereas anatomists utilize radioligands to map the regional localization of proteins[14]. However, the limited spatial resolution and the incompatibility with general immunolabeling approaches prevent the usefulness of autoradiography for nanoscale molecular imaging in cellular and subcellular target profiles.

Already more than 3000 human proteins are known to be "druggable"[15,16] and extensive efforts are underway to identify new and more selective drug–target combinations[17–19]. Yet, fluorescent drugs are not routinely considered as alternative or complementary tools to antibodies or radioligands for the nanoscale anatomical localization of molecular interactions[20]. Although previous studies have applied fluorescent ligands in different modalities of optical microscopy[21–28], the pertinence of labeled small molecules for nanoscale pharmacological analysis that can be interpreted within the native cellular and subcellular context of complex tissue samples by using a single-molecule localization microscopy-based approach remains unexplored.

In this study, we aimed to establish a method that benefits from the advantages of using fluorescent pharmacoprobes together with single-molecule localization microscopy, while bypassing the disadvantages of antibody-based immunolabeling and radioligand-based autoradiography. We coined the term "PharmacoSTORM" to indicate that a combination of the rational molecular design of fluorophore-tagged pharmacologically active compounds with the exceptional single-molecule detection sensitivity of Stochastic Optical Reconstruction Microscopy (STORM)[4–6,19] was exploited to determine target engagement of drugs in physiologically relevant subcellular, cellular and tissue contexts. We demonstrate the broad applicability of PharmacoSTORM for quantitative ligand-binding measurements for major protein families (G protein-coupled receptors, ion channels, and enzymes), and for cariprazine, an antipsychotic and antidepressant drug. Nanoscale pharmacology has never been applied to complex tissue preparations and has not been used to identify in vivo target engagement sites in cell- and subcellular compartment-specific manner yet[20]. Therefore, as a proof-of-principle, we sought to exploit PharmacoSTORM to determine the specific binding sites of cariprazine, a rapidly emerging new medication (Vraylar™/Reagila™) applied for the treatment of bipolar disorder and schizophrenia. We discovered that in vivo-administered cariprazine predominantly binds to $D_3$ dopamine receptors distributed along the axons of granule cells in the hilar subregion of the Islands of Calleja and avoids dopaminergic afferent varicosities in the mouse brain. These findings illuminate the substantial psychiatric importance of a poorly understood brain region in the ventral forebrain and provide unexpected insights into the neurobiological mechanisms underlying the therapeutic effects of cariprazine.

## Results

**Versatility of the PharmacoSTORM approach.** The increasing number of high-resolution protein structures together with robust computational approaches, such as in silico binding predictions and improved synthesis methodologies[18,19,29] culminated in a rapidly evolving understanding of the molecular recognition process and the drug–target interactions formed. Therefore, we hypothesized that rational selection and design of fluorescently labeled drugs can successfully extend the available set of probes amenable for super-resolution imaging.

First, we applied a generalizable workflow for the production and validation of a PharmacoSTORM probe for one of the most abundant G protein-coupled receptors in the brain, the $CB_1$ cannabinoid receptor ($CB_1R$) (Fig. 1a). After concurrent analysis of the known high-affinity ligands and the recently revealed receptor structure[30], we selected a hydroxy-cannabinol core suitable for chemical modifications, optimized the site and linker for Sulfo-Cy5 fluorophore attachment by in silico docking experiments, and synthesized a fluorescent $CB_1R$ ligand (ABP511, Supplementary Information, Fig. 1a, b, and Supplementary Fig. S1). Next, we characterized the high affinity ($K_i$ = 3.5 nM, Supplementary Table S1) and potent $CB_1$ agonism of the fluo-cannabinoid pharmacoprobe in ligand-binding and functional assays (Fig. 1a). Fluo-cannabinoid binding exhibited remarkable selectivity for $CB_1R$, and untransfected HEK 293 cells were devoid of fluo-cannabinoid binding. Moreover, visualization of fluo-cannabinoid binding sites could be combined with immunolabeling for $CB_1$ receptors (Fig. 1a, c and Supplementary Fig. S2a). Importantly, the fluorescence intensities derived from the two labeling modalities were highly correlated, demonstrating the quantitative power of fluo-cannabinoid imaging (Supplementary Fig. S2b–d). Next, we performed correlated confocal microscopy and STORM super-resolution imaging[31]. Confocal

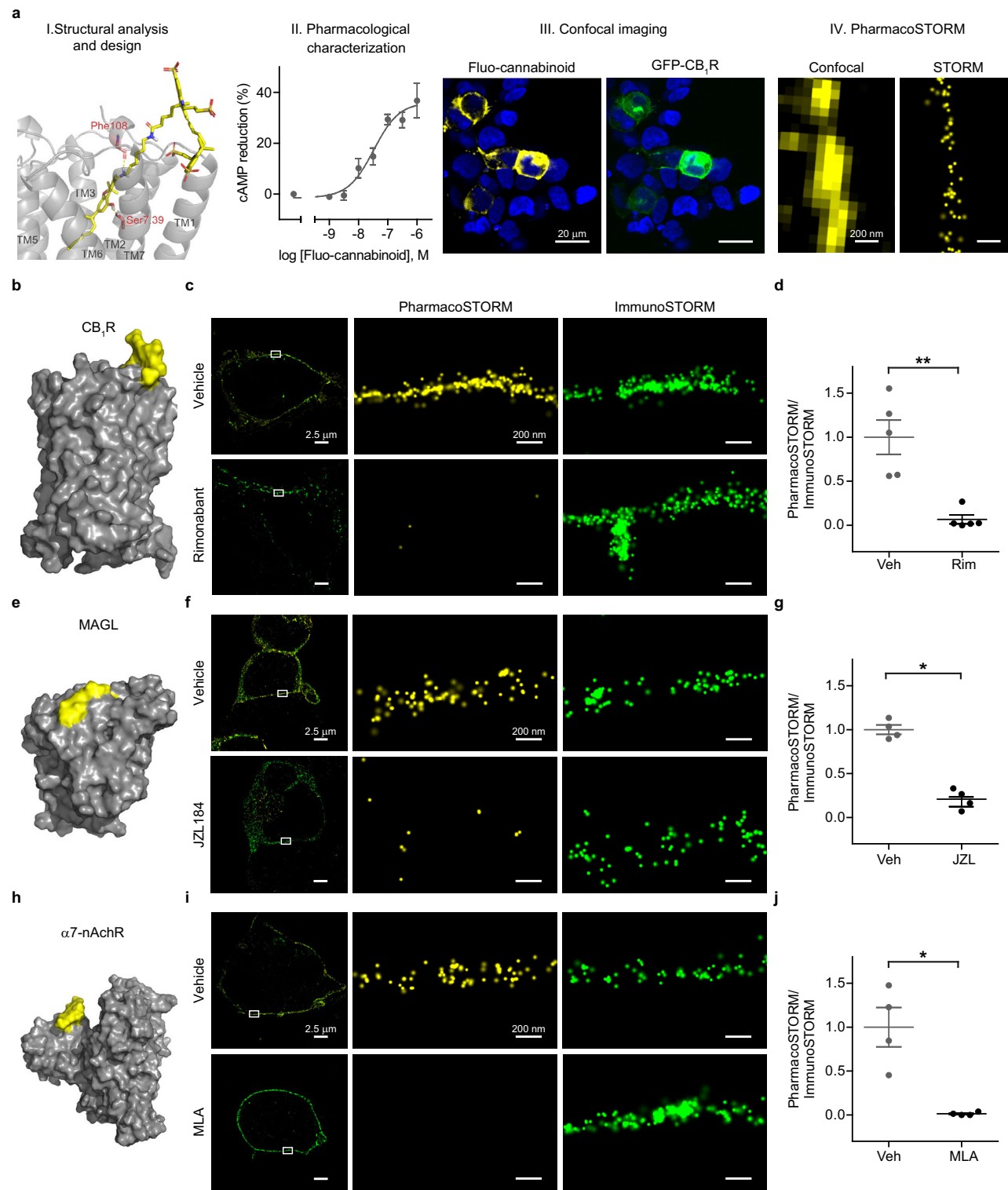

imaging showed the plasma membrane with a diffraction-limited resolution (Fig. 1a). In contrast, the binding of fluo-cannabinoid molecules to $CB_1R$ was detected with nanoscale precision by PharmacoSTORM imaging that enabled the visualization of the diffraction-unlimited outline of the plasma membrane (Fig. 1a, c). We took advantage of classical competitive ligand-binding measurements to demonstrate the binding specificity of the fluo-cannabinoid pharmacoprobe for $CB_1Rs$. Pretreatment with the unlabeled $CB_1R$ antagonist, rimonabant completely prevented fluo-cannabinoid binding (Supplementary Fig. S2a and Fig. 1c, d)

and fully eliminated PharmacoSTORM localization points representing $CB_1Rs$, while the antibody-based $CB_1R$ Immuno-STORM signal remained unaffected (Fig. 1c, d).

In another set of experiments, we aimed to test the broad applicability of PharmacoSTORM by investigating other major protein families. We designed and characterized a fluorescent activity-based probe (DH-463) for monoacylglycerol lipase (MAGL) (Supplementary Fig. S3 and Supplementary Information), a serine hydrolase enzyme that is the master regulator of endocannabinoid and prostaglandin levels in the brain[32,33]. The

**Fig. 1 Quantitative nanoscale imaging of major protein families by PharmacoSTORM. a** General workflow of the development of PharmacoSTORM probes, demonstrated by the example of a $CB_1R$ ligand. (I) The docking pose of the fluorescently tagged $CB_1R$ agonist (fluo-cannabinoid, yellow) in the $CB_1R$ (gray) (PDB:5XRA). The hydrogen bonds between the protein residues (red) and the probe are represented as yellow dashed lines. (II) In vitro cAMP assay for the functional characterization of fluo-cannabinoid activity on $CB_1R$ ($n = 5$, $logEC_{50} = -7.48$ nM, $E_{max} = 36.1\%$). Reference compound was WIN55,212-2 ($n = 4$, $logEC_{50} = -8.03$ nM, $E_{max} = 51.5\%$). (III) Confocal microscopic image of HEK 293 cells expressing $GFP-CB_1R$, labeled with 100 nM fluo-cannabinoid; (IV) Comparison of confocal and STORM images depicting fluo-cannabinoid binding to $CB_1Rs$ on the same plasma membrane segment. **b, e, h** Surface representations of three PharmacoSTORM probes, bound to their respective protein targets that represent three major protein families. **b** GPCR:$CB_1R$, (PDB:5XRA, gray) with fluo-cannabinoid (yellow); **e** enzyme: MAGL (PDB:6BQ0, gray) with DH-463 (yellow); **h** ion channel: $\alpha$7-nAchR (PDB:4HQP, gray) with fluorescent $\alpha$-bungarotoxin (yellow). **c, f, i** Competitive ligand-binding measurements by dual direct STORM imaging demonstrate the high specificity of PharmacoSTORM probes. Target proteins were visualized simultaneously by pharmacoprobe- and antibody-based labeling. **c** 100 nM fluo-cannabinoid and anti-GFP immunostaining for $GFP-CB_1R$ in the absence (vehicle) or presence of $CB_1R$ antagonist rimonabant (Rim, 10 μM). **f** 100 nM DH-463 and anti-GFP immunostaining for GFP-MAGL pretreated either with vehicle or the MAGL inhibitor JZL184 (JZL, 10 μM). **i** 1 ng/μl Alexa647-$\alpha$-bungarotoxin and anti-HA immunostaining for $\alpha$7-nAchR-HA pretreated either with vehicle or the $\alpha$7-nAchR antagonist methyllycaconitine (MLA, 10 μM). **d, g, j** Scatter dot plots display the ratio of PharmacoSTORM and ImmunoSTORM localization points (LPs) in the absence or presence of the unlabeled competitive ligands. Data are normalized to average vehicle (Veh) pretreatment. Two-tailed Mann–Whitney tests were performed. **d** $n = 5$, $P = 0.0079$; **g** $n = 4$, $P = 0.0286$; **j** $n = 4$, $P = 0.0286$. Data are presented as mean ± SEM, $n$ values indicate biologically independent experiments in all panels.

Cy5-conjugated covalent inhibitor of MAGL (Fig. 1e and Supplementary Fig. S2e–h) could also efficiently bind to the enzyme protein and helped to uncover the nanoscale distribution of the enzyme in PharmacoSTORM experiments (Fig. 1f). Competitive ligand-binding measurements with an unlabeled MAGL inhibitor (JZL184) verified the high binding selectivity of the fluorescent substrate for MAGL and confirmed the specificity of its PharmacoSTORM signal (Fig. 1f, g). Ion channels represent the third major class of proteins that are often very difficult to label by antibodies due to their embedding into large macromolecular complexes. In light of its increasing psychiatric significance[34] and based on the availability of a high-affinity labeled ligand, we chose the $\alpha$7-nicotinic acetylcholine receptor ($\alpha$7-nAChR). Alexa647-tagged $\alpha$-bungarotoxin selectively labeled $\alpha$7-nAChR and visualized its nanoscale distribution on the surface of HEK 293 cells with the help of PharmacoSTORM imaging (Supplementary Fig. S2i–l and Fig. 1h, i). Pretreatment with an $\alpha$7-nAChR-specific antagonist (methyllycaconitine, MLA) fully prevented the appearance of the PharmacoSTORM signal but did not affect the ImmunoSTORM signal (Fig. 1j and Supplementary Fig. S2i). Collectively, these findings indicate that PharmacoSTORM is a versatile and efficient approach to visualize ligand binding at the nanoscale level.

**Nanoscale pharmacological characterization of fluo-cariprazine by PharmacoSTORM.** Given its broad applicability, we reasoned that PharmacoSTORM may also provide important insights into the pharmacological mechanism of action for clinically used drugs. We focused on a centrally acting compound because the tremendous complexity of the brain poses exceptional difficulties for pharmaceutical development. Cariprazine is a medicine for the treatment of schizophrenia and for both the manic and depressive episodes of bipolar I disorder[35–37]. Interestingly, this recently approved drug displays a unique pharmacological profile among the third generation antipsychotics, because it exhibits the highest affinity toward $D_3$ dopamine receptors ($D_3R$) and shows a magnitude of order weaker binding to the much more common $D_2$ receptors ($D_2R$)[38]. However, despite its rapidly increasing therapeutic use and broadening indications, the neurobiological mechanism of its action has remained rather elusive. Moreover, selective antibodies for $D_3R$ are currently unavailable, hence fluo-cariprazine can serve as a unique labeling tool for this enigmatic dopamine receptor. Therefore, we noted that the generation of a fluorescent cariprazine derivative is important, and we hypothesized that the nanoscale characterization of its target engagement sites would strongly facilitate the limited mechanistic understanding of the remarkable therapeutic effectiveness of cariprazine.

By applying rational molecular design, we synthesized a Sulfo-Cy5-tagged cariprazine analog (ABP535; Fig. 2a, Supplementary Fig. S4 and Supplementary Information). Radioligand measurements verified that fluo-cariprazine preserved the high affinity and preference towards $D_3R$ ($K_i = 1.31$ nM, Supplementary Table S1). Pharmacological experiments established that fluo-cariprazine behaves as a weak partial agonist similarly to cariprazine (Fig. 2b, c). Moreover, fluo-cariprazine signal was exclusively detected in HA-tagged $D_3R$-expressing cells, but not in untransfected HEK 293 cells. In addition, the pharmacoprobe signal strongly correlated with the expressed receptor protein levels and the fluorescent intensity of pharmacoprobe labeling reliably followed the receptor distribution inhomogeneities in the plasma membrane (Fig. 2d and Supplementary Fig. S5). Notably, fluo-cariprazine-binding sites displayed a lateral localization precision comparable to the Cy5 fluorophore localization precision (~7 nm), when measured from the distribution of individual fluorophore blinking events during STORM imaging (Supplementary Fig. S5e). This observation verifies the intact photo-chemical properties of cariprazine-attached Sulfo-Cy5. Competitive PharmacoSTORM ligand-binding measurements using the $D_3R$ selective antagonist, SB277011-A, further corroborated strong fluo-cariprazine-binding specificity to $D_3R$ (Fig. 2e, f).

The preference of fluo-cariprazine towards $D_3R$ over $D_2R$, the classical target of antipsychotics, was also investigated at the structural and the nanoscale levels. In line with our molecular dynamics simulations using the recently published cryo-EM receptor structures[39,40], fluo-cariprazine PharmacoSTORM signal was significantly reduced in $D_2R$-expressing cells (Supplementary Fig. S6). To further challenge our structure-guided approach, we first predicted that phenylalanine 338 (F338) plays an essential role in maintaining high-affinity $D_3R$ conformation, then we integrated PharmacoSTORM into a site-directed mutagenesis study. In accordance with the simulation, we detected strongly reduced binding of fluo-cariprazine to F338A-mutated $D_3R$ (Supplementary Fig. S7). To investigate the quantitative power of the PharmacoSTORM approach, a plasma membrane-delimited saturation binding assay was carried out with fluo-cariprazine in live cells and its binding to $D_3Rs$ exhibited a classical sigmoidal response function when measured by PharmacoSTORM (Fig. 2g). Remarkably, fluo-cariprazine binding could readily be detected even at sub-nanomolar ligand concentration due to the exceptional single-molecule detection sensitivity of STORM imaging (Fig. 2g–i). Since a single $D_3R$

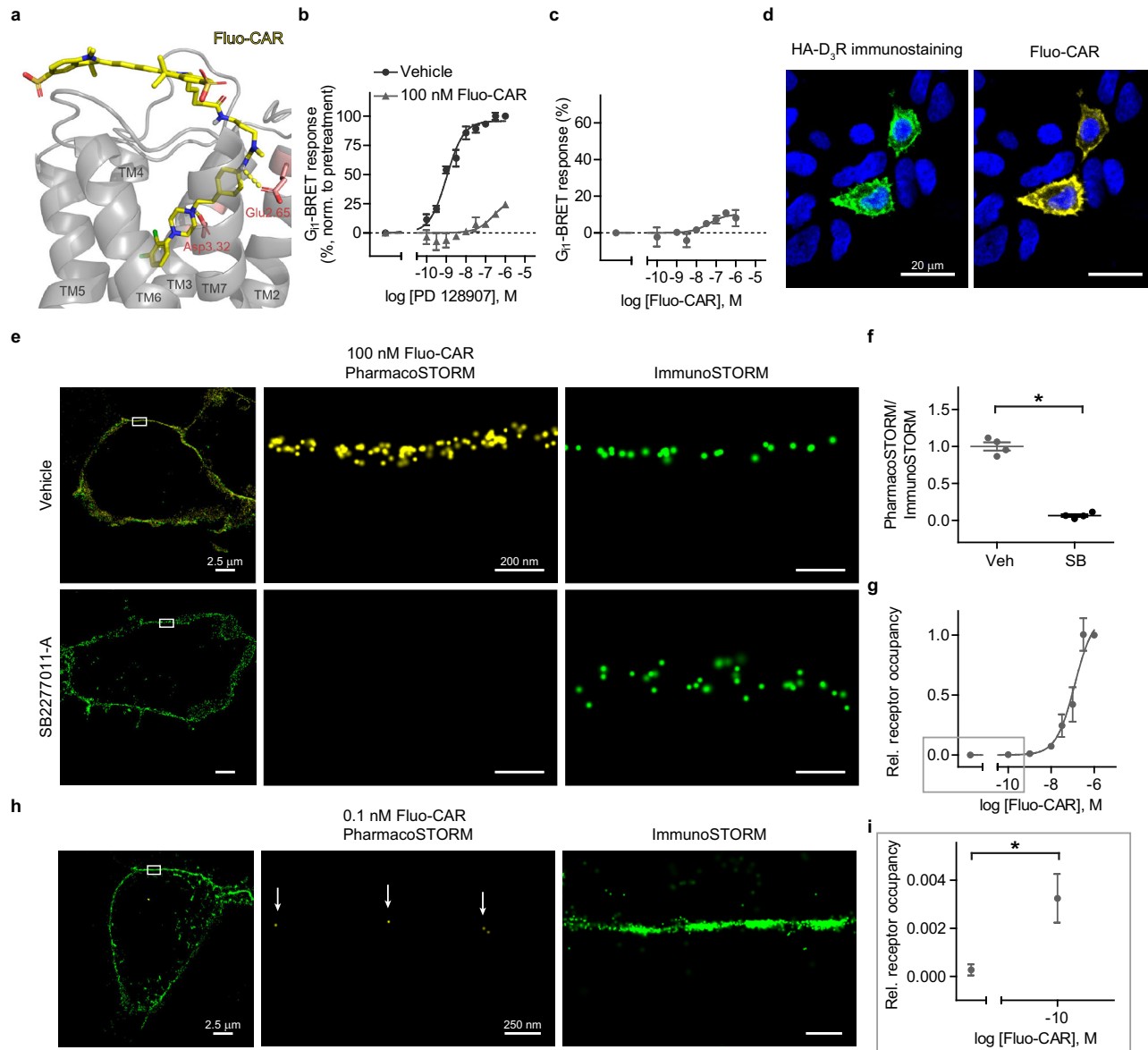

**Fig. 2 Development and nanoscale pharmacological characterization of fluo-cariprazine. a** The docking pose of the fluorescently tagged cariprazine (Fluo-CAR, yellow) in a $D_3R$ (gray) homology model (for details, see Supplementary Information). The hydrogen bonds between the key protein residues (Glu2.65, Asp3.32 in red) and the pharmacoprobe are represented as yellow dashed lines. **b, c** Functional effects of Fluo-CAR on $D_3R$ in BRET-based $G_{i1}$-activation assay. **b** Fluo-CAR antagonized the effect of the $D_3R$ selective agonist PD128907. Cells were incubated with Fluo-CAR or vehicle for 5 min before PD128907 application. Fluo-CAR pretreatment increased the $EC_{50}$ value of PD128907 from 1.047 nM to 256.1 nM ($n = 3$). **c** Fluo-CAR alone acted as a weak partial agonist. Sigmoidal concentration-response curve was fitted ($EC_{50} = 44.1$ nm). BRET ratios are expressed as the percentage of the signal of 1 µM PD128907 ($n = 3$). **d** Representative confocal images of HA-$D_3R$-expressing HEK 293 cells treated with Fluo-CAR (100 nM) following HA immunostaining. Fluo-CAR selectively binds those cells that express HA-$D_3R$. **e, f** Competitive ligand-binding experiment assessed by quantitative dual direct STORM imaging. HA-$D_3R$-expressing cells were incubated with either vehicle (Veh) or 10 µM SB277011-A (SB), a selective $D_3R$ antagonist. The ratio of PharmacoSTORM and ImmunoSTORM localization points (LPs) was normalized to vehicle pretreatment. SB277011-A pretreatment markedly reduced the Fluo-CAR signal ($n = 4$, two-tailed Mann–Whitney $U$ test, $P = 0.0286$). **g** Saturation binding assay of Fluo-CAR treatment measured within the plasma membrane. Relative (rel.) receptor occupancy (the ratio of PharmacoSTORM and ImmunoSTORM LPs) was expressed as the percentage of the signal of 1 µM Fluo-CAR, $n$ values are (from left to the right) 4; 4; 3; 4; 3; 3; 3; 7. One-site sigmoidal binding curve was fitted (half-maximal occupancy was at 123 nM). **h, i** Nanoscale visualization of Fluo-CAR binding sites in the sub-nanomolar concentration range. The representative image is shown in (**h**). White arrows point to sparsely bound individual Fluo-CAR molecules. **i** Higher magnification of the lower concentration range from (**g**) is shown (two-tailed Mann–Whitney $U$ test, $P = 0.0294$). Data are presented as mean ± SEM, $n$ values indicate biologically independent experiments in all panels.

protein binds a sole fluo-cariprazine molecule, which is equipped with only one fluorophore tag, an individual drug–receptor interaction was often detected as a single STORM blinking event in the plasma membrane. Our systematic analysis revealed that this precise stoichiometry makes PharmacoSTORM probes especially advantageous tools for true single-molecule localization

studies (Supplementary Fig. S8). Notably, 78% of fluo-cariprazine molecules were detected as a single STORM localization point. On the other hand, commercially available secondary antibodies carry unknown number of fluorophores. Accordingly, only 36% of antibodies were represented as a single blinking event, and the rest were detected as "localization point clusters" with variable

numbers of members. These measurements suggest that the application of fluorescent pharmacoprobes makes STORM data amenable for more precise quantitative measurements.

We also tested the feasibility of PharmacoSTORM-based ligand-binding measurements in identified compartments of cells with complex morphology, such as neurons. Fluo-cariprazine specifically labeled primary hippocampal neurons transfected with HA-D$_3$Rs, and PharmacoSTORM-based visualization of D$_3$Rs in the neuronal plasma membrane could be readily combined with consecutive immunolabeling for MAP2 protein, a marker of dendrites (Supplementary Fig. S9).

Taken together, these findings showcase the broad application potential of PharmacoSTORM to study clinically relevant compounds by exploiting its quantitative power and spatial localization precision for pharmacological measurements at the nanoscale level in living cells.

**PharmacoSTORM-based multiscale anatomical framework identifies the Islands of Calleja as a major cariprazine-binding site in the brain**. We reasoned that PharmacoSTORM and fluorescent pharmacoprobes can be as useful for the visualization of the regional distribution of drug binding as autoradiography but have the unique capacity to zoom beyond the diffraction limit and to detect single drug molecules at the nanoscale level even within complex tissue preparations. To illustrate the feasibility of multiscale anatomical mapping, we next mapped fluo-cariprazine binding in the mouse brain by the combination of epi-fluorescence, confocal and PharmacoSTORM super-resolution imaging.

Acute brain slices were prepared according to the well-established protocol of electrophysiological experiments and were incubated with fluo-cariprazine in vitro. After the treatment, the slices were fixed, re-sectioned, and further processed for different microscopic techniques ("Methods"). Brain slices from knockout mice lacking D$_3$Rs and in vitro pharmacological pretreatment with unlabeled cariprazine were used in control experiments. The highest abundance of dopaminergic fibers is known to be concentrated in the striatum[41]. Low-magnification broad topological examination of the striatal areas revealed the condensed accumulation of fluo-cariprazine-binding sites in the ventral part of the forebrain that is known as the tubular striatum[42]. This binding pattern was in striking contrast to the very weak fluo-cariprazine-binding density throughout the dorsal and ventral striatum (Fig. 3a and Supplementary Fig. S10a). Higher magnification uncovered that the vast majority of fluo-cariprazine binds to several small areas in the vicinity of dense cell populations representing the so-called Islands of Calleja (Fig. 3a and Supplementary Fig. S10a). Although this region was described in the late 1800s, its functional significance has remained poorly understood[43]. The lack of selective neurochemical markers resulted in controversial anatomical descriptions and it is still debated whether these cells belong to several independent nuclei or form a single brain structure[43–45]. Remarkably, the high density of fluo-cariprazine binding helped to reveal that the cell masses, which appear as separate islands on coronal sections, are connected via continuous "bridge-like" structures within the hilar subregion outlined by fluo-cariprazine binding (Fig. 3b). Complete three-dimensional reconstruction of fluo-cariprazine-rich areas confirmed that the entire complex extends over an unexpectedly large area as a single brain structure almost at the scale of the adjacent nucleus accumbens (Fig. 3c–e).

Considering the high affinity of fluo-cariprazine for D$_3$Rs (Fig. 2), we tested the hypothesis whether the high density of D$_3$Rs in the Islands of Calleja accounts for the prominent fluo-cariprazine binding in the hilar subregion. Indeed, fluo-

cariprazine labeling was absent in the Islands of Calleja of D$_3$R knockout mice (Fig. 3f, g). To investigate the feasibility of quantitative and specific PharmacoSTORM measurements in brain tissue, we repeated the comparison at the nanoscale level (Fig. 3h, i and Supplementary Fig. S10b, c, e). Notably, PharmacoSTORM imaging could confirm the robust decrease in the number of STORM localization points in D$_3$R knockout animals (Fig. 3h, i and Supplementary Fig. S10e) that could not be reduced further by unlabeled cariprazine pretreatment indicating that D$_3$ dopamine receptors represent the sole binding sites of fluo-cariprazine in the Islands of Calleja (Supplementary Fig. S10d, e). In contrast, PharmacoSTORM imaging successfully detected the low copy number of D$_3$Rs in the adjacent ventral striatum, but not in the dorsal striatum, where D$_3$ receptors are known to be either present or absent, respectively[46] (Supplementary Fig. S10b, c, f, g). In accordance with the very high density of D$_2$ dopamine receptors in the dorsal and ventral striatum[47], and in agreement with the lower affinity of cariprazine and fluo-cariprazine for D$_2$Rs compared to D$_3$Rs (Supplementary Fig. S6), PharmacoSTORM quantification established ~10 times less fluo-cariprazine-binding density in both the ventral and dorsal striatum that could be eliminated by unlabeled cariprazine pretreatment (Supplementary Fig. S10b, c, f, g).

We also tested whether the classical autoradiography protocols that are widely used in numerous pharmacological laboratories would be amenable for PharmacoSTORM imaging. Therefore, we repeated fluo-cariprazine measurements on cryosections obtained from fresh-frozen mouse brain tissue. We found the same regional distribution of fluo-cariprazine-binding sites in the Islands of Calleja (Supplementary Fig. S11a), and the lack of the PharmacoSTORM signal in D$_3$R knockout animals (Supplementary Fig. S11b, c). However, we found that the quality of the general ultrastructure of fresh-frozen tissue was inferior compared to live tissue preparations. Therefore, we performed the subsequent set of imaging experiments on fluo-cariprazine-treated acute living brain slices widely used by electrophysiologists that were re-sliced after fixation to sections applicable for conventional anatomical approaches.

Finally, STORM localization points that represent the precise sites of drug–target interactions were registered with high localization precision (median of 9.4 nm at 5 μm tissue depth) (Fig. 3j). In striking contrast, confocal microscopic images did not allow the differentiation of single drug molecules from background or to detect their anatomical organization. In conclusion, these results demonstrate the broad potential of fluorescent pharmacoprobes and PharmacoSTORM imaging for multiscale visualization of ligand binding from gross anatomical levels down to fine nanoscale levels within complex tissues.

**Nanoscale visualization of in vivo drug–target interactions in the Islands of Calleja**. Detailed anatomical information on the precise cellular and subcellular sites of therapeutic action as well as on additional drug-binding sites that accounts for potential side effects are both fundamentally important for mechanism-driven pharmacotherapy. We, therefore, investigated whether nanoscale molecular imaging by PharmacoSTORM can be used in combination with immunolabeling that is capable to outline well-defined anatomical structures and provides the relevant regional, cellular, and subcellular contexts for the drug–target interaction sites. Based on the representative PharmacoSTORM images, the distribution of fluo-cariprazine-binding sites gave the impression that the drug targets are widely scattered within the hilar subregion of the Islands of Calleja. To quantitatively analyze the nanoscale distribution of fluo-cariprazine-binding sites, we compared nearest-neighbor distance measurements between fluo-

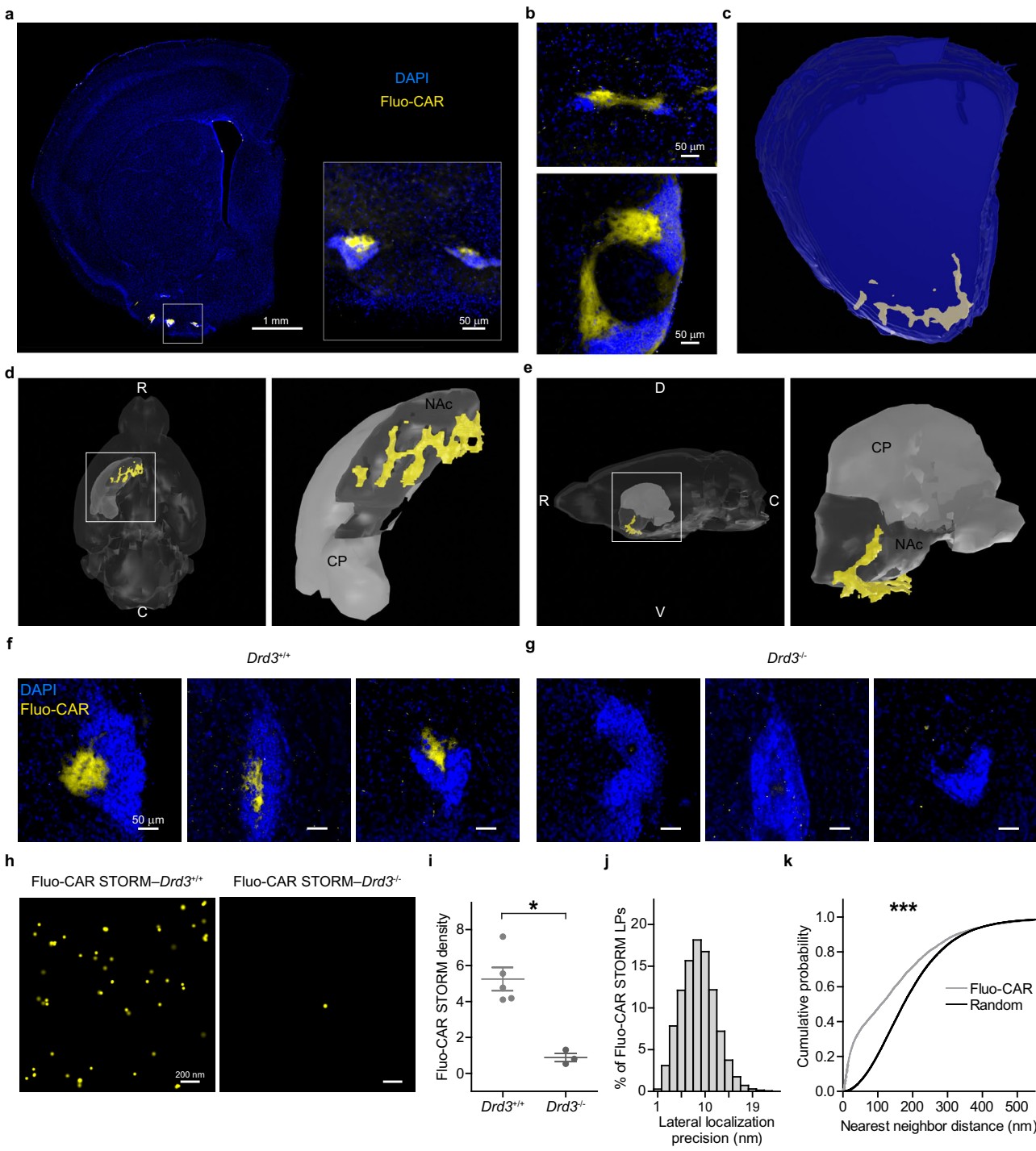

cariprazine localization points and randomized datasets (Fig. 3k). We found that fluo-cariprazine molecules accumulated significantly more neighbors at small nanoscale distances than it would be expected from a random distribution. These data suggest that fluo-cariprazine-binding sites are preferentially associated with certain subcellular structures in the hilar subregion, and the detection sensitivity and localization precision of STORM imaging may provide a means to unravel the underlying morphological structures and the nanoscale architecture of dopaminergic signaling organization in these anatomical contexts.

We, therefore, visualized functionally related dopaminergic signaling proteins together with fluo-cariprazine binding. Surprisingly, the level of dopamine- and cAMP-regulated neuronal phosphoprotein (DARPP-32), a key molecule of intracellular dopaminergic signaling in striatal neurons was below the detection threshold throughout the fluo-cariprazine-rich areas (Fig. 4a). In contrast, a dense meshwork of tyrosine hydroxylase (TH)-immunopositive dopaminergic afferents was observed in the same hilar subregion, where fluo-cariprazine-binding sites were concentrated (Fig. 4b). Dopamine transporter (DAT)-immunolabeling followed the same distribution pattern as TH at both the regional and nanoscale levels, verifying an extensive dopaminergic innervation of the Islands of Calleja (Supplementary Fig. S12).

A growing body of evidence indicates that synaptic and non-synaptic dopaminergic signaling encode distinct neurobiological

**Fig. 3 Multiscale anatomical mapping of fluo-cariprazine-binding sites in the brain. a** Low-magnification epifluorescence image of a mouse brain coronal section after incubation with 300 nM Fluo-CAR (yellow) and nuclear staining with DAPI (blue). The inset highlights the intense Fluo-CAR labeling in the Islands of Calleja. **b** Although the high-density cell masses of the Islands of Calleja appear distinctly segregated on coronal sections, specific areas that are rich in Fluo-CAR labeling interconnect them. These Fluo-CAR-positive areas represent the hilus of the Islands of Calleja. **c–e** 3D morphological analysis of the gross anatomical architecture of the Islands of Calleja based on prominent Fluo-CAR labeling. Precise reconstruction of high-density binding sites (**c**) demonstrates that the hilus represents a large, continuous area. Ventral (**d**) and lateral (**e**) views of the reconstruction (yellow), integrated into a 3D mouse brain atlas (dark gray: nucleus accumbens (NAc), light gray: caudoputamen (CP)). White capital letters mark anatomical directions (R rostral, C caudal, V ventral, D dorsal). **f, g** Representative images of Fluo-CAR labeling in the Islands of Calleja from $Drd3^{+/+}$ (**f**) and $Drd3^{-/-}$ (**g**) animals. Acute slices were treated with 300 nM Fluo-CAR. Fluo-CAR binding is dramatically decreased throughout the entire brain structure in the absence of $D_3$Rs. **h** Demonstration of the feasibility of PharmacoSTORM super-resolution imaging in brain tissue preparations. Images were taken in the hilus of the Islands of Calleja of $Drd3^{+/+}$ and $Drd3^{-/-}$ animals after 300 nM Fluo-CAR treatment. **i** Two-tailed Mann–Whitney $U$ test was performed ($P = 0.0357$) to compare Fluo-CAR binding site density (STORM LPs/$\mu m^2$) in the hilus of $Drd3^{+/+}$ ($n = 5$) and $Drd3^{-/-}$ mice ($n = 3$). **j** Lateral (x-y) localization precision of Fluo-CAR binding sites based on the analysis of STORM LP blinking distribution. The median lateral localization precision was 9.42 nm ($n = 98,260$ LPs). **k** Quantitative analysis of the nanoscale distribution of Fluo-CAR-binding sites within the hilus. Nearest-neighbor distance measurements were performed between Fluo-CAR STORM LPs and between the equal numbers of randomly distributed STORM LPs. Cumulative distribution functions were statistically compared with Kolmogorov–Smirnov test ($P < 0.001$) and revealed non-random tissue distribution.

information via activating $D_1$ and $D_2$ dopamine receptors on striatal medium spiny neurons[48,49]. How $D_3$ receptors contribute to dopaminergic neuromodulation in a synaptic and/or non-synaptic manner remained largely unexplored. After identification of the Islands of Calleja as a major brain circuit controlled by $D_3$R-mediated signaling, we next aimed to analyze the nanoscale relationship of dopamine release sites and $D_3$R distribution. At the diffraction-limited light microscopic level, there was a considerable overlap between fluo-cariprazine signal and tyrosine hydroxylase (TH)-positive dopaminergic varicosities. However, correlated confocal and PharmacoSTORM imaging and quantitative analysis of drug-binding site distribution demonstrated that fluo-cariprazine rich areas do not overlap with dopaminergic axon terminals (Fig. 4c, d). Thus, in contrast to the apparent overlap in confocal microscopic images, the nanoscopic mismatch indicates that $D_3$Rs are not concentrated as auto-receptors on dopaminergic varicosities. As a next step, we determined whether there is a precisely ordered nanoarchitecture between the dopaminergic varicosities and dopamine receptors similarly to the so-called nanocolumns described in glutamatergic and GABAergic synapses[50]. Dual-color PharmacoSTORM and ImmunoSTORM imaging with fluo-cariprazine and TH-immunolabeling, respectively, revealed that the nanoscale distribution of drug-binding sites is independent of the relative localization to dopaminergic nerve terminals (Fig. 4e, g). The unexpected lack of a strict nanoscale association implicates that the vast majority of $D_3$Rs are not intrasynaptic receptors and may instead play a major role in dopaminergic volume transmission.

Next, we asked the question whether unlabeled cariprazine administered to live mice exhibits a similar in vivo nanoscale distribution as we observed in vitro by using fluo-cariprazine. This experiment also addressed the methodical issue whether nanoscale anatomical information about the in vivo ligand-binding pattern of a drug applied at a known dose can be inferred from PharmacoSTORM imaging experiments based on the displacement of its respective fluorescent pharmacoprobe. Because cariprazine crosses the blood-brain barrier as a centrally acting drug, intraperitoneal injection of either vehicle or unlabeled cariprazine (1 mg/kg) was administered to live mice. Acute brain slices were prepared 2 h later for pharmacological labeling with fluo-cariprazine (Fig. 5a). As a proof-of-concept of the competition for binding sites, unlabeled cariprazine completely eliminated the fluo-cariprazine signal in the hilar subregion of the Islands of Calleja (Fig. 5b). We also demonstrated that the competitive binding experiment can be analyzed by a combination of PharmacoSTORM imaging of fluo-cariprazine distribution and confocal imaging of TH-

immunopositive dopaminergic nerve terminals. By exploiting the quantitative power of PharmacoSTORM, we measured the number of localization points representing fluo-cariprazine-binding sites. Notably, a dramatic decrease upon in vivo cariprazine pretreatment was found compared to vehicle treatment (Fig. 5c, d). In accordance with the previous in vitro data, we did not observe fluo-cariprazine-binding sites on dopaminergic varicosities in the vehicle-treated animals indicating that dopamine may reach the high-affinity $D_3$Rs via non-synaptic transmission (Figs. 4c, d and 5e). In addition, these data provide compelling evidence that the high density of fluo-cariprazine PharmacoSTORM signal in the Islands of Calleja directly corresponds to real in vivo target engagement sites of cariprazine. Because the approach is easily generalizable, pharmacoprobe-based PharmacoSTORM super-resolution imaging can thus be an ideal method to provide high-precision ligand-binding data of numerous unlabeled drugs in complex tissues.

**PharmacoSTORM imaging of the cell-type- and subcellular compartment-specific nanoscale distribution of cariprazine-binding sites.** Given the absence of fluo-cariprazine binding to dopaminergic terminals, we aimed to determine the precise site of cariprazine action within the Islands of Calleja circuitry in a cell-type- and subcellular domain-specific manner. To this end, we developed a methodology that combines physiological and morphological measurements with the registration of nanoscale molecular ligand-binding data obtained from the very same neuron.

Previous studies of the Islands of Calleja cytoarchitecture have shown that granule cells represent its principal cell type[43,51,52], although there is limited knowledge about their functional or morphological characteristics. Patch-clamp electrophysiological recordings were used to label individual neurons via electrodes containing biocytin for morphological reconstruction and subsequently treated with fluo-cariprazine in acutely prepared brain slices. Correlated three-dimensional analysis of neuronal morphology and fluo-cariprazine distribution revealed that granule cells have two major process types. One process enters the hilar subregion and its branches remain strictly confined to the same area where high fluo-cariprazine binding is concentrated (Fig. 6a, b and Supplementary Fig. S13). The other process preferentially arborizes amongst the cell bodies within the granular subregion (Fig. 6a, b and Supplementary Fig. S13). To identify the morphological types of these neurites that can predict the physiological role of $D_3$R-mediated dopamine signaling, we performed electron microscopic analysis of the ultrastructure of biocytin-filled Islands of Calleja granule cells. Biocytin-containing

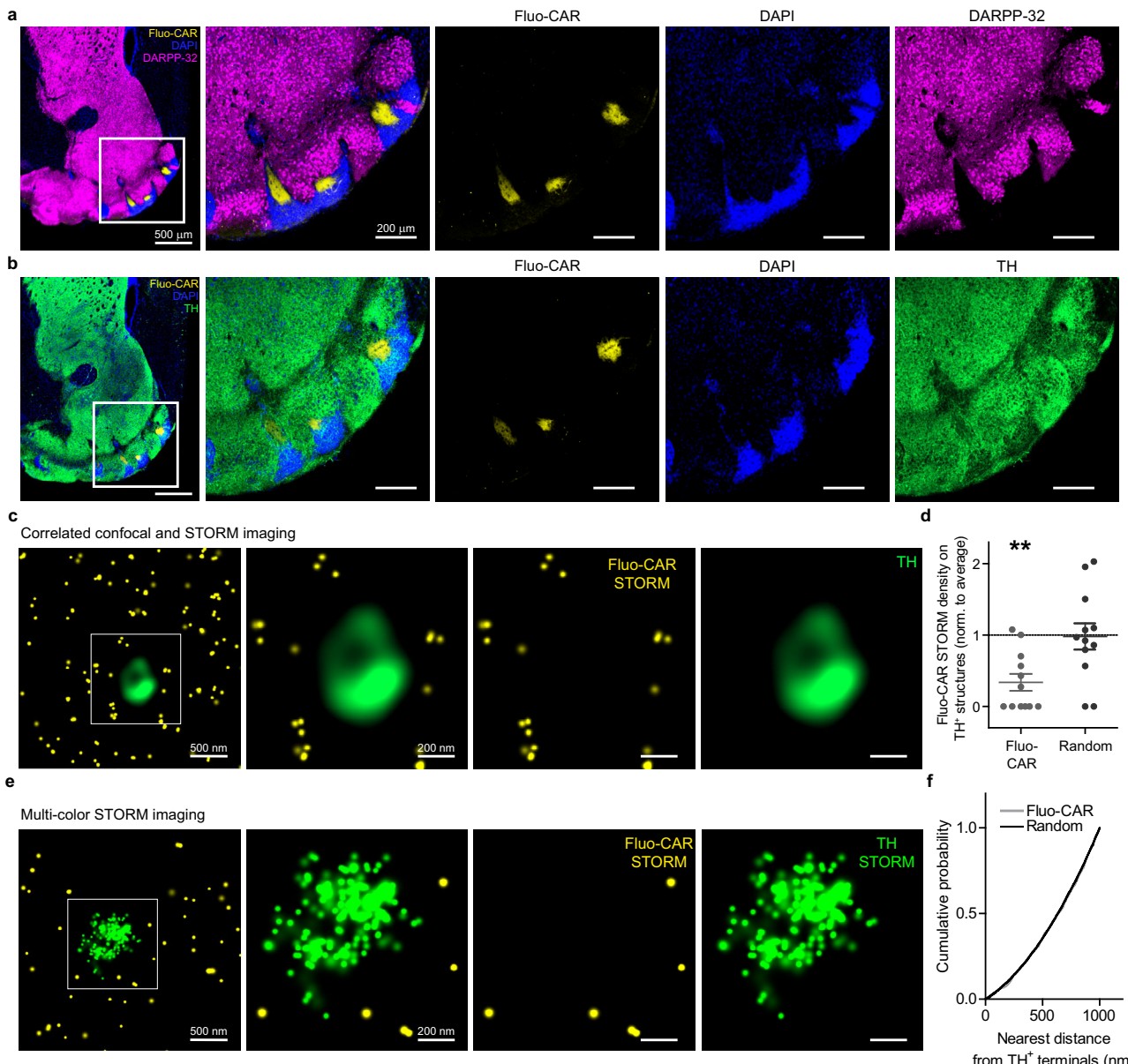

**Fig. 4 Correlated PharmacoSTORM molecular and anatomical imaging visualizes Fluo-CAR binding surrounding dopaminergic nerve terminals. a, b**
Comparative analysis of the distribution of fluo-cariprazine-binding sites and well-known dopaminergic signaling proteins in the ventral part of the mouse forebrain. **a** Fluo-CAR-treated slices (300 nM) were immunostained for DARPP-32. Neither the hilar subregion (outlined by yellow Fluo-CAR binding) nor the cell bodies in the granular subregion (visualized by blue DAPI staining) of the Islands of Calleja contain the striatal marker protein DARPP-32 (purple).
**b** In contrast to the characteristic DARPP-32 immunonegativity, immunostaining for tyrosine hydroxylase (TH) in the Islands of Calleja suggests that dopaminergic axons densely innervate this brain region. **c** Correlated confocal and PharmacoSTORM imaging of TH-containing-varicosities and Fluo-CAR binding, respectively. Although the dopaminergic varicosities are surrounded by a high density of Fluo-CAR binding sites, Fluo-CAR apparently avoids TH-immunopositive dopaminergic nerve terminals. **d** Quantitative nanoscale spatial analysis of Fluo-CAR binding in relation to dopaminergic afferents of the Islands of Calleja. Data are normalized to the average density of PharmacoSTORM localization points (LPs) on each image. Note that if the same number of Fluo-CAR LPs would be randomly distributed then TH-containing varicosities would bear the average density on each image. Two-tailed Wilcoxon signed-rank test was used to test if median is different from 1 ($n = 12$, data are from five animals, presented as mean ± SEM; $P = 0.0048$ for Fluo-CAR LPs, $P = 0.7836$ for random distribution). **e** Dual-color PharmacoSTORM and ImmunoSTORM imaging of Fluor-CAR labeling and TH-immunostaining. **f** Random nanoscale distribution of Fluo-CAR-binding sites around dopaminergic axon terminals. To test the nanoscale relationship between the site of dopamine synthesis and dopamine binding sites, the nearest-neighbor distance of each Fluo-CAR-binding site was measured from a 2D convex hull fitted onto the TH ImmunoSTORM LPs. The identical cumulative distribution of Fluo-CAR LPs and randomized LPs (Kolmogorov–Smirnov test, two-tailed $P = 0.8749$) indicates the lack of specific Fluo-CAR binding site nanoclusters in the proximity of TH+ boutons as one would expect in case of synaptic specializations and rather suggests volume transmission as the primary mode of dopaminergic signaling.

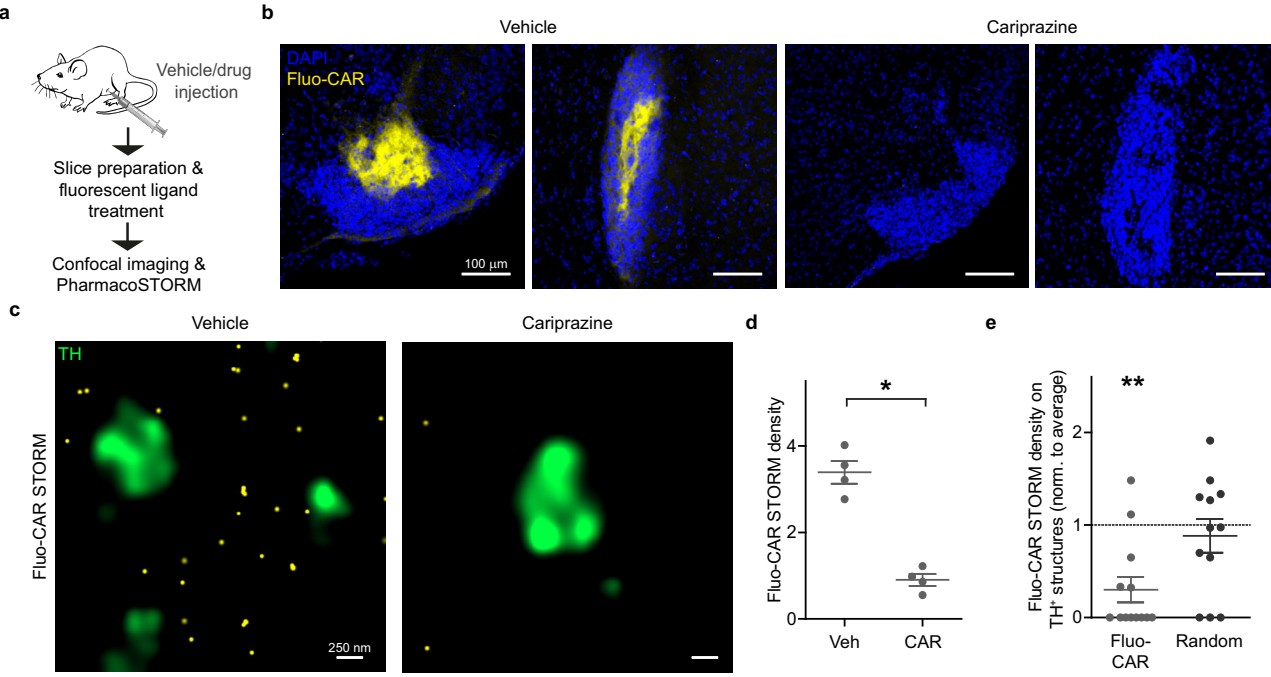

**Fig. 5 Nanoscale visualization of in vivo cariprazine drug interaction sites by PharmacoSTORM. a** Schematic illustration of the generalizable workflow that enables PharmacoSTORM-based quantitative nanoscale analysis of target engagement sites of drugs administered in vivo. In the specific experiment, intraperitoneal injection of 1 mg/kg cariprazine was followed 2 h later by acute brain slice preparation and incubation with 30 nM Fluo-CAR. **b** Confocal images show that in vivo cariprazine pretreatment efficiently blocks Fluo-CAR binding in the hilus of the Islands of Calleja. **c** Fluo-CAR binding sites (yellow) are visualized by PharmacoSTORM imaging. Note the striking lack of Fluo-CAR-binding sites on TH-immunopositive nerve terminals. In unlabeled cariprazine-treated animals, the density of STORM LPs representing Fluo-CAR-binding sites was markedly reduced, indicating that Fluo-CAR binding sites are preoccupied with cariprazine administered in vivo. **d** Scatter dot plot shows PharmacoSTORM density (LPs/$\mu m^2$) in the Islands of Calleja in cariprazine- or vehicle-treated animal pairs ($n = 4$). Data are presented as mean ± SEM. Each data point represents the average density of STORM images obtained in the Islands of Calleja from each animal. Statistical evaluation was performed by two-tailed Mann–Whitney $U$ test ($n = 4$, $P = 0.0286$). **e** Statistical analysis of the nanoscale distribution of Fluo-CAR STORM LPs in relation to TH-immunopositive varicosities. The same analysis was applied as in Fig. 4d. The number of Fluo-CAR STORM LPs was significantly lower on TH-containing structures ($n = 12$ frames, $n = 4$ animals). Two-tailed Wilcoxon signed-rank test was performed to test if the median of data is different from the hypothetical median of 1 ($P = 0.0057$ for Fluo-CAR LPs, $P = 0.5042$ for random LP distribution).

profiles in the granular subregion showed characteristic structural features of dendrites based on their afferent synaptic specialization. In contrast, hilar processes showed the hallmarks of axons with varicosities representing bouton-like structures filled with synaptic vesicles (Fig. 6c–h). These electron microscopic images also confirmed that the general ultrastructure of the granule cells is well preserved in the acute brain slice preparation during biocytin-filling that was similar to the fluo-cariprazine treatment protocol (Fig. 6c–h).

We finally performed PharmacoSTORM experiments on electrophysiologically and morphologically characterized granule cells to investigate whether the dense fluo-cariprazine signal in the hilar subregion is due to the direct binding of fluo-cariprazine to the axons of Islands of Calleja granule cells (Fig. 7). In contrast to the lack of fluo-cariprazine-binding sites on dopaminergic terminals, the STORM localization points representing $D_3Rs$ followed the outline of the biocytin-filled structures and displayed specific nanoscale segregation on the hilar axons of the Islands of Calleja granule cells. Quantitative analysis of the prominent PharmacoSTORM signal density on these subcellular compartments statistically verified this observation (Fig. 7g). Notably, drug-binding sites were present on both axon terminals and on interconnecting preterminal axons (Fig. 7f–h). Since the same drug–receptor interaction can hold substantially different functional consequences depending on the subcellular compartment-specific localization, we further analyzed whether fluo-cariprazine displays biased nanoscale distribution at distinct subcellular

compartments of the axon (Fig. 7i–j). However, Pharmaco-STORM localization point density was not significantly different between the axon terminals and the preterminal segments (Fig. 7j) also indicating a more general role of dopaminergic volume transmission in the control of the efferent output of granule cells.

These findings highlight the functional significance of the Islands of Calleja granule cell axons in the hilar subregion as a central site of action of cariprazine thereby paving the way for a better understanding of the neurobiological mechanisms of cariprazine's therapeutic effects.

## Discussion

In this study, we introduce a general approach for the application of pharmacological probes to localize proteins at the nanoscale level and developed a pipeline to visualize drug–target interactions in a cell- and subcellular compartment-specific manner in complex tissue preparations. Given its broad applicability, as demonstrated on various protein families (GPCRs, enzymes, and ion channels), and the concentrated efforts of the pharmacological community to develop probes for every protein[53,54], it is conceivable to predict that the PharmacoSTORM approach will be advantageous for numerous research purposes in life sciences. As a proof-of-concept example, the PharmacoSTORM approach was instrumental to discover a specific binding site of cariprazine, an emerging psychiatric drug at an anatomical location within a largely understudied brain circuit.

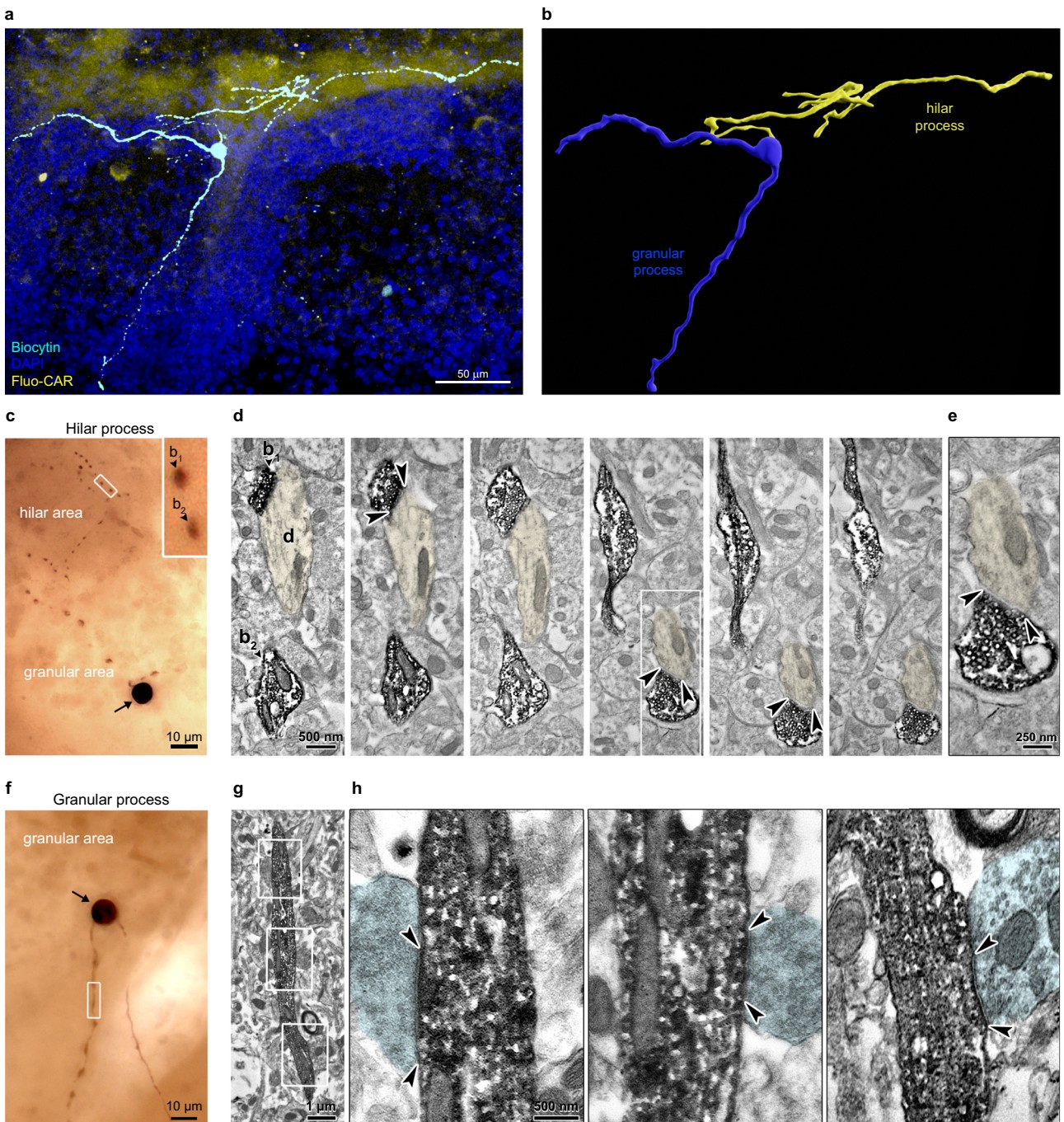

**Fig. 6 Morphological properties of granule cells in the Islands of Calleja. a, b** A representative granule cell is shown in the Islands of Calleja. This neuron was filled with biocytin during whole-cell patch-clamp recording in the acute coronal slice preparation. The same slice was live-stained with 300 nM Fluo-CAR for the correlated visualization of drug-binding sites and the morphological features of the granule cell. Note that the arborization of the granule cell strictly follows the unique shape of the Islands of Calleja. One of its processes preferentially branches in the hilar subregion, whereas two other major processes arborize in the granular subregion. **b** 3D reconstruction of the same granule cell. The neuronal process that enters the Fluo-CAR-rich hilar subregion is marked in yellow and hereinafter referred to as the hilar process. The other neurites that remain restricted to the granular subregion are termed the granular processes (blue). **c–h** Correlated light and electron microscopy was used to determine the integrity of the acute slice preparations used for PharmacoSTORM staining and helped to determine the morphological features of hilar and granular processes. **c** A representative light micrograph of a biocytin-filled granule cell soma (arrow) in the granular subregion of the Islands of Calleja and its hilar process that has numerous varicosities within the hilus. Inset shows two bouton-like structures ($b_1$, $b_2$) at higher magnification from the boxed area. **d** Electron microscopic images of consecutive serial sections demonstrate that the two boutons ($b_1$, $b_2$) are filled with synaptic vesicles, and form symmetric synaptic specializations (arrowheads) on a dendrite (**d**, pseudo-colored in yellow). **e** High-magnification electron micrograph of the synapse from the boxed area in (**d**). **f** Another biocytin-filled granule cell (arrow) is shown together with its granular process. **g** Electron microscopic image of the boxed area in (**f**). **h** High-magnification electronmicrographs of boxed areas in (**g**), showing afferent boutons (pseudo-colored in cyan) forming chemical synapses (arrowheads) on the granular process of the granule cell. Panels display representative micrographs from three biologically independent experiments.

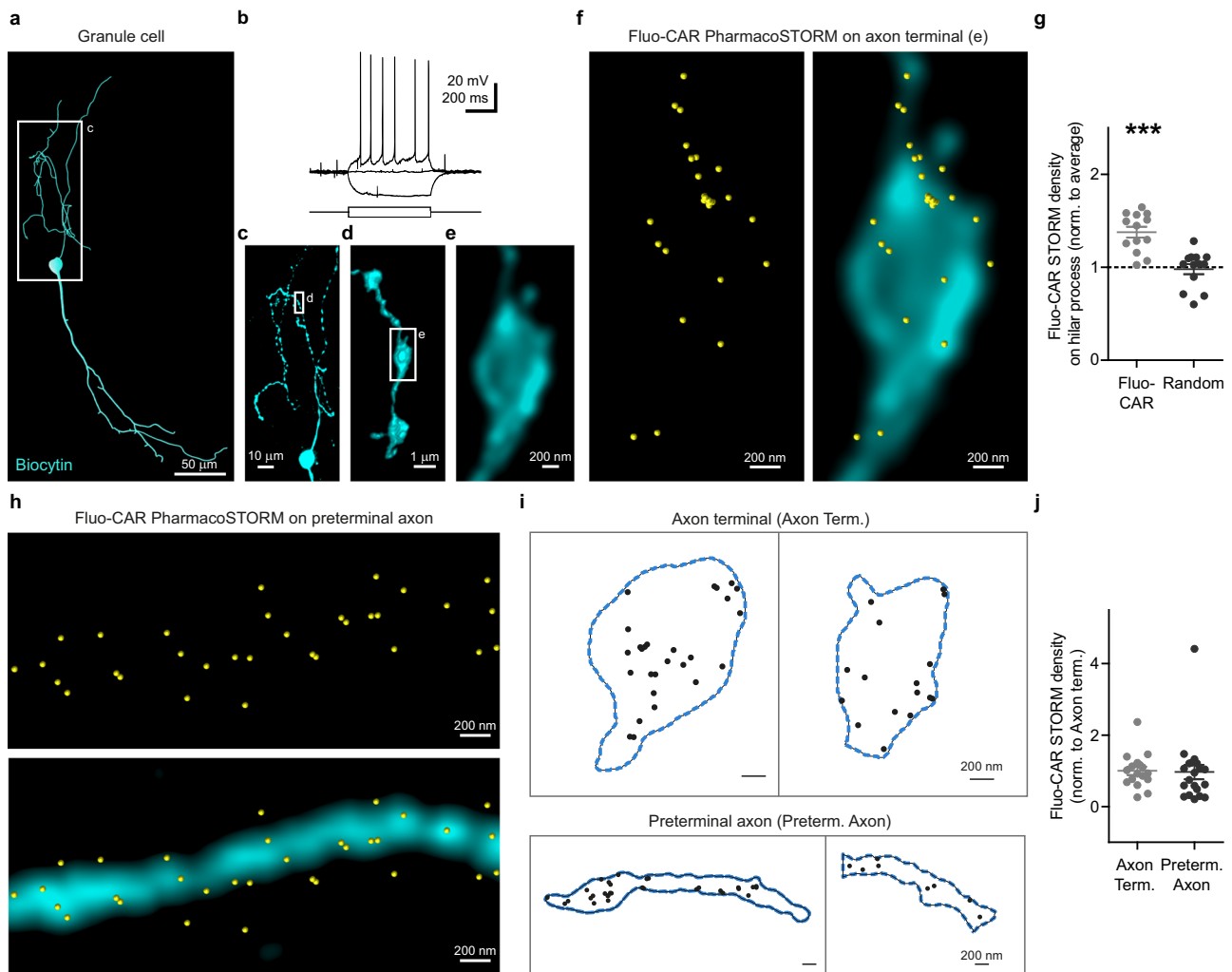

**Fig. 7 Cell-type- and subcellular compartment-specific nanoscale distribution of fluo-cariprazine binding on the axons of the granule cells in the Islands of Calleja. a–f** Combination of nanoscale molecular imaging of Fluo-CAR-binding sites with cellular anatomical and electrophysiological characterization. **a** Neurolucida reconstruction of a representative biocytin-filled granule cell in the Islands of Calleja. **b** Voltage traces in response to +7 pA, 0 pA, −10 pA current steps from resting membrane potential recorded in whole-cell current-clamp configuration reveal the action potential firing pattern of the same granule cell shown in (**a**). **c** Maximum intensity z-projection of the confocal image stack of the hilar process in the boxed area in (**a**). **d** Volume view of a high-resolution confocal image stack taken from a varicose segment from boxed area in (**c**). **e** Higher magnification deconvolved confocal image of the boxed area in (**d**) illustrates a single axon terminal of the granule cell. **f** Correlated confocal and PharmacoSTORM imaging of the granule cell bouton (cyan) presented in (**e**), and the corresponding Fluo-CAR-binding sites (yellow) along the surface of the axon terminal. **g** Scatter dot plot shows the LP density on granule cell axons normalized to the average LP density on the same image. Two-tailed Wilcoxon signed-rank test shows that Fluo-CAR density on the axon of biocytin-filled cells is significantly higher than expected from random distribution (n = 13, n = 6 animals, P = 0.0002; P = 0.8394 for randomly distributed LPs). **h** Correlated confocal and PharmacoSTORM image of Fluo-CAR binding sites (yellow) on a long preterminal segment of a granule cell axon (cyan). **i** Schematic illustration of representative segmented PharmacoSTORM images for further analysis. Contours of axon terminals and preterminal axon segments are shown as blue dashed lines. Fluo-CAR LPs are represented as black dots. **j** Comparative analysis of nanoscale Fluo-CAR-binding site density on the distinct axonal subcompartments. Values are normalized to the mean density on axon terminals. Two-tailed Mann–Whitney test revealed no significant difference of Fluo-CAR-binding site density between the two subcellular compartments (axon terminals: n = 17, preterminal axon segments: n = 20, P = 0.402).

The design of sensitive and specific fluorescent pharmacoprobes can be challenging, yet, the emerging field of structural biology offers powerful tools for the rational design of high-affinity pharmacoprobes[18,19,29] (Supplementary Information). Functional pharmacological assays are convenient for the rapid screening of the best candidates, as we demonstrated via the development of a fluorescent ligand for the CB$_1$ cannabinoid receptor, the MAGL enzyme and a labeled analog of an approved drug (fluo-cariprazine). The exact effects of fluorophore tagging on equilibrium or kinetic ligand-binding parameters can certainly be obtained by further radioactivity- or fluorescence-based

methods[55,56]. Moreover, the specificity of target binding can be analyzed by appropriate control PharmacoSTORM experiments using untransfected cells, displacement assays, point-mutant receptors for binding sites, or tissues obtained from knockout mice as we demonstrated all four control approaches in the case of fluo-cariprazine.

The application of fluorescent ligands may introduce multiple technical improvements to super-resolution imaging[20] and expedite microscopic developments in the future that were previously hindered by certain practical limitations of immunolabeling. Labeled pharmacoprobes may ease the direct

quantification of single-molecule localization data due to the known stoichiometry of target engagement and the constant number of fluorophores on each probe. Although the variability of the blinking events/dyes remains a factor that can still add variance to PharmacoSTORM data but it can be controlled during data analysis and we also provided evidence that single fluo-cariprazine molecules bound to $D_3Rs$ are predominantly detected as a single localization point. In contrast, the application of regularly used antibodies, which carry a variable number of fluorophores, may easily lead to the overcounting of their targets without strict validation[12].

It is also important to consider the potential advantages of fluorescent ligands for live-cell imaging. Immunolabeling is known to cause artificial clusters that are prone to misinterpretation[57,58]. On the other hand, ligand-induced protein dimerization/oligomerization, or receptor internalization can hold important biological information, and single-molecule localization microscopy has been shown to be an excellent way to investigate these processes[23,24]. Also, the fast on-rate kinetics of small molecules is especially beneficial in live-cell super-resolution experiments[23], however, only pharmacoprobes with a low dissociation rate are well suited for sustained imaging of their targets. The systematically optimized labeling and sample handling protocols described in this study ("Methods" and Supplementary Information) can substantially extend the potential imaging period and are also instrumental for the combination of PharmacoSTORM imaging with post hoc immunolabeling. The length and frequency of washing steps after pharmacoprobe incubation during tissue sample preparations reduce nonspecific binding of the labeling probes but also limit the visualization of target sites with higher dissociation rates. Thus, their investigation may require modified protocols to overcome the time constraints and to decrease the dissociation of the probe. However, recent breakthrough developments in covalent ligands is also expected to expand the toolset amenable for PharmacoSTORM imaging[59]. Accordingly, we showed the usefulness of an irreversible enzyme inhibitor. Given the high signal-to-noise ratio achieved with the activity-based probe, DH-463, other fluorescently labeled activity-based probes are also predicted to be ideal tools for high-precision studies of enzymatic reactions.

We show that fluorescent ligand-based labeling can be combined with immunolabeling for dual-color STORM imaging in cell culture and tissue experiments. Recent advances in multi-color STORM imaging of tissue samples have paved the way for the simultaneous nanoscale analysis of numerous proteins[60]. Our results with fluo-cariprazine suggest that important clinical drug targets (such as $D_3R$) may be henceforth included even in the absence of a selective antibody. Moreover, the rapid and homogeneous diffusion of pharmacoprobes within live tissue preparations represents a clear advantage over immunolabeling of chemically fixed samples in which antibody penetration is often inhomogeneous and incomplete. We were able to demonstrate the high efficiency of pharmacoprobe diffusion in acute brain slices, since fluo-cariprazine binding could also be investigated throughout a 2.5 mm-thick slice after an hour of incubation. Thus, it is conceivable that PharmacoSTORM imaging will also benefit from the exploitation of the recent developments of deep tissue volumetric single-molecule localization microscopy approaches in the future[61–64].

Certainly, fluorescent ligands also require rigorous validation. However, the specificity of pharmacoprobes can be easily validated by displacement assays with selective unlabeled compounds, without the demand for expensive or unavailable knockout animals that are needed for the validation of a new antibody. Accordingly, all molecules described in this study were validated by in vitro competitive PharmacoSTORM ligand-binding displacement measurements both in cell cultures and in tissue samples. Moreover, we established an experimental workflow in which unlabeled drugs acting in vivo can displace pharmacoprobes within the tissue. As an example, we demonstrated that cariprazine administration to live mice effectively blocks fluo-cariprazine binding in the Islands of Calleja. This means that the detected nanoscale fluo-cariprazine signal represents a real subset of the in vivo targets of the original cariprazine drug. A major advantage of this approach is that a single well-characterized PharmacoSTORM compound is sufficient to provide quantitative in vivo data about the high-precision localization of any therapeutics or excipients that occupy the same target in the tissue. In this regard, fluo-cariprazine will be of high interest, because the $D_3R$ has been recently identified as an in vitro off-target of certain excipients, but receptor occupancy in tissue after systemic exposure remained poorly defined[65].

Radioligand binding has traditionally been measured using tissue- and cell culture membrane preparations or analyzed by low-resolution autoradiographic maps[14]. Therefore, the cell-type- or compartment-specific characteristics of pharmacological interactions remained hidden. Here, we show that the PharmacoSTORM approach can overcome these limitations and provide data about the multi-level distribution of drug-binding sites. We introduced a series of technical improvements that enabled PharmacoSTORM to visualize drug molecules from regional up to the nanoscale level in combination with commonly used fluorescent staining methods in tissue. To illustrate these advantages, we directly visualized fluo-cariprazine-binding sites together with immunolabeling of dopaminergic axons and on electrophysiologically and morphologically characterized biocytin-filled neurons.

Large-scale analysis and 3D reconstruction of ligand-binding sites revealed the preference of fluo-cariprazine toward a large continuous neuronal assembly, that is embedded in the tubular striatum[42] and corresponds to the anatomical structure of the Islands of Calleja. Previous studies, that attempted to discern the complex morphology of this brain region often relied solely on nuclear staining or the increased NADPH-diaphorase reactivity of this region[43–45]. However, we found that the Islands of Calleja cannot be reliably distinguished on fluorescent images from adjacent striatopallidal regions unless specific markers are applied. As an example, the troughs of the Islands of Calleja that cross multiple layers of the olfactory tubercle part of the tubular striatum and reach the pial surface can be easily mixed with the dense cap compartments or ruffled regions of the olfactory tubercle, which has also led to confusion during its first anatomical descriptions[43]. On the other hand, the application of fluo-cariprazine labeling helped to define its exact localization and to visualize the precise boundaries of the hilar area. The present anatomical data unequivocally prove the integrity of this brain region, which has been controversial before[43–45].

The abundant fluo-cariprazine binding completely disappeared in the Islands of Calleja in $D_3R$ KO animals and could be displaced by in vitro and in vivo-administered cariprazine. These findings suggest that $D_3Rs$ in this region mediate important behavioral effects of this recently approved medicine. Despite earlier low-resolution in situ hybridization and autoradiography experiments that suggested the presence of $D_3Rs$[46,66,67], surprisingly, the functional involvement of the Islands of Calleja in the action of antipsychotics and antidepressants has not yet been experimentally demonstrated. Certainly, $D_3Rs$ are also likely to be present in other brain regions[68–72], albeit at much lower concentration. By relying on the unique single-molecule detection sensitivity of PharmacoSTORM, we could visualize low copy numbers of $D_3Rs$ in the ventral striatum. Furthermore, additional cariprazine-binding sites, presumably $D_2Rs$, could also be

detected in both the dorsal and ventral striatum at one order of magnitude lower binding density. Thus, in future studies, the in vivo competitive ligand-binding assay together with cell-type-specific knockout mice will also be useful to identify the precise site of given behavioral actions of cariprazine on additional molecular and cellular targets in different brain circuits that may also contribute to the therapeutic and adverse effects of cariprazine[38].

$D_3R$ possesses higher affinity to dopamine than other dopamine receptor subtypes[73] which raises the possibility, that dopamine activates $D_3R$ at low tissue concentrations and relatively far from its release site[73,74]. Accordingly, correlated PharmacoSTORM and ImmunoSTORM imaging of fluo-cariprazine target sites and dopaminergic nerve terminals clearly showed the lack of nanoscale association of $D_3Rs$ with the site of dopamine synthesis. This implicates that $D_3Rs$ are predominantly involved in dopaminergic volume transmission. On the other hand, cell-type-specific PharmacoSTORM imaging revealed the preferential binding of fluo-cariprazine molecules to the axon of granule cells, which represent the principal neuron type in the Islands of Calleja implicating the important role of dopaminergic signaling and $D_3Rs$ in shaping neuronal efferent outputs from this region. The presence of high-affinity $D_3Rs$ on granule cell axons also implies that the output of these neurons is controlled by transient alterations of dopamine concentration encoding salient events associated with negative affective values[75]. The impaired functionality of emotional salience is also a hallmark of negative symptoms in patients with schizophrenia spectrum disorders. Therefore, the identification of high-density binding sites in the Islands of Calleja for cariprazine that displays uniquely high efficacy in the treatment of negative symptoms is clinically highly relevant[35,36,46]. In addition, the widespread role of $D_3Rs$ in other psychiatric disorders[71,73,76] and the emerging therapeutic significance of cariprazine in the treatment of other psychiatric conditions[35–37,46], further emphasizes that the strategic nanoscale localization of in vivo cariprazine binding discovered by PharmacoSTORM imaging, is expected to draw major attention to the clinical importance of Islands of Calleja granule cells.

## Methods
Methods of chemical synthesis of probes is discussed in the Supplementary Information section. Supplementary Tables contain detailed information about oligo primer sequences (Supplementary Table S2), antibodies (Supplementary Table S3), and commercial ligands (Supplementary Table S4).

**Plasmid DNA constructs**. The coding sequence of human *DRD3* gene was subcloned from Addgene #66270 plasmid into pcDNA3.1 vector with hDRD3_fw and hDRD3_rev primers between HindIII and EcoRV restriction sites. To generate N-terminally HA-tagged $D_3R$, we amplified the coding sequence of $D_3R$ with HA-hD3_fw and hDRD3_rev primers, and the PCR product was inserted into pcDNA3.1 vector between HindIII and BamHI restriction sites. The Phe338Ala mutation was introduced to HA-tagged $D_3R$ by precise gene fusion PCR using F338-fw and F338-rev primers. N-terminally HA and C-terminally Venus-tagged $D_3R$ was generated by cloning DRD3 coding sequence from Addgene #66270 with HA-hD3_fw and hDRD3_N1_rev primers.

C-terminally Venus-tagged $D_3R$ was generated by cloning human *DRD3* coding sequence from Addgene#66270 plasmid with primers: hDRD3_N1_fw and hDRD3_N1_rev into empty N1-mVenus plasmid. To generate $D_2R$-Venus, human *DRD2* coding sequence was amplified from Addgene #66269 with hDRD2_fw and hDRD2_N1_rev primers into empty N1-mVenus plasmid. Generation of N-terminally EGFP-tagged mouse $CB_1R$ was previously described[8]. The coding sequence of monoacylglycerol lipase (MAGL) was amplified with mMGL_fw and mMGL_rev primers using a mouse brain cDNA library as a template. The PCR product was digested with HindIII and BamHI enzymes and cloned into pcDNA3.1 backbone. GFP-MGL was constructed by subcloning the open reading frame of MAGL into pAcGFP1-C1 vector following HindIII/BamHI digestion. To generate C-terminally HA-tagged human α7-nAChR, we used plasmid #62276 from Addgene as a template for PCR with CHRNA7_HA_fw and CHRNA7_HA_rev primers. The amplified DNA was inserted into pcDNA3.1 vector between the BamHI and XhoI restriction sites. TMEM35 in pCMV6 vector was from Origene (RC209790). The pGloSensor™-22F cAMP plasmid was from Promega. The

plasmid encoding $CB_1R$ was generated by inserting the coding sequence of human $CB_1R$ into pCAGIG vector. The plasmids encoding $G_{i1}$ subunits ($α_{i1}$, $β_1$, and $γ_2$) were from the cDNA Resource Center. We created $Gα_{i1}$–Rluc8 using overlap exchange PCR, Rluc8 was inserted into $Gα_{i1}$ between residue 91 and 92 with SGGGGS linkers similar to a previous design[77]. Venus–$β_1$ was created by replacing YFP with Venus in YFP–$β_1$ using restriction digestion, YFP–$β_1$ plasmid construct was described previously[78].

**Cell culture, transfection, in vitro drug treatment**. HEK 293 and HEK 293T cells were obtained from American Type Culture Collection (ATCC CRL-1573 and CRL-3216 respectively). Cells were cultured in Dulbecco's Modified Eagle Medium supplemented with 10% fetal bovine serum (Biosera) at 37 °C in 5% $CO_2$ and were tested for potential mycoplasma contamination on a monthly basis.

For microscopic experiments, cells were seeded one day before transfection on poly-D-lysine-coated coverslips in 24-well culture plates. Coverslips were cleaned with extensive 2 M NaOH and 70% ethanol washing steps prior to use. Transient transfection was done with Lipofectamine 2000 (Thermo Fischer) according to the manufacturer's protocol. After transfection (20–28 h), pharmacological treatment was applied and then combined with immunolabeling. The full procedure was optimized for each investigated protein. In the case of $D_3R$-, $CB_1R$-, and α7-nAChR-expressing cells, all labeling and washing steps were executed in Hanks' balanced salt solution (Sigma-Aldrich H6648) on live cells, and in Dulbecco's Phosphate Buffered Saline (Corning, 21-031-CV) on fixed cells. The labeling steps with antibodies, ligands, or fixatives were followed by three washing steps. In competitive binding experiments, cells were pretreated with a solution containing the desired concentration of the unmodified drug or equal volume of its vehicle (DMSO for SB277011, rimonabant, and JZL184, and distilled water for MLA). Unless otherwise stated, we used CF568-labeled secondary antibodies.

The HA-$D_3R$-expressing cells were live-stained with anti-HA primary and secondary antibodies for 10 min at room temperature. In the case of competitive binding experiments, cells were treated with 10 μM SB277011 (Tocris) or vehicle (DMSO) for 5 min before 20-min fluo-cariprazine incubation. Fluo-cariprazine treatment and all subsequent steps were done at 4 °C. Cells were fixed in 4% PFA for 10 min before imaging. For Supplementary Fig. S5g, Cy5-labeled secondary antibody was applied, without fluorescent ligand treatment. For fluo-cannabinoid binding experiments, all steps were executed at room temperature. After anti-GFP immunolabeling, cells were incubated with rimonabant or DMSO for 5 min, then with fluo-cannabinoid for 1 min, and fixed before imaging. MAGL pharmacological labeling was performed in culture medium in a 5% $CO_2$ atmosphere at 37 °C. JZL184 or vehicle (DMSO) pretreatment, as well as DH-463 treatment lasted 2 h. Cells were fixed with 4% PFA for 10 min. Before immunolabeling, cells were washed with 100% methanol three times for 10 min and were blocked with 5% normal donkey serum (NDS) in 0.1 M phosphate buffer (PB; pH 7.4), for 30 min. Anti-GFP immunolabeling was done in 0.1 M PB and immunostained cells were postfixed with 4% PFA before imaging. In the case of α7-nAChR experiments, its chaperon protein, NACHO (TMEM35) was co-transfected in a 10 to 1 receptor:chaperon ratio in order to enhance cell-surface expression of the ion channel[79]. MLA pretreatment and Alexa647-bungarotoxin labeling were done at 4 °C for 20 min. After pharmacological labeling, the cells were fixed with 4% PFA, and subsequent anti-HA-immunolabeling was done at room temperature. Immunolabeling was followed by post-fixation. To characterize individual fluorescent drug molecules and secondary antibodies in the plasma membrane, the results of the following two experimental setups were compared: sparse fluo-cariprazine labeling was investigated by treating HA-$D_3R$ or HA-$D_3R$-Venus expressing cells as in the previously described experiments, but fluo-cariprazine was applied at 0.1 nM. Secondary antibodies were investigated after treating live HA-$D_3R$-Venus-expressing cells with a regular concentration of anti-HA primary antibodies, but with low concentration (0.001 μg/ml) Alexa647-conjugated secondary antibodies.

### Functional in vitro assays
*cAMP measurement*. $CB_1R$ activation was assessed by measuring the inhibition of forskolin-induced cAMP accumulation. HEK 293T cells were transfected with plasmids encoding pGloSensor™-22F cAMP sensor, human $CB_1R$ and $G_{i1}$ ($α_{i1}$, $β_1$, and $γ_2$ subunits) in suspension and plated on poly-D-lysine-coated white 96-well plates (Greiner 655083). After transfection (20 h), cells were pre-equilibrated for 2 h at room temperature with pGloSensor™-22F cAMP reagent according to the manufacturer's protocol. Luminescence was measured with a Thermo Scientific Varioskan Flash multimode plate reader. After baseline luminescence measurement, cells were incubated with increasing concentrations of fluo-cannabinoid or WIN55,212-2 for 10 min. Thereafter, 2.5 μM forskoline was added to increase the intracellular cAMP level, and luminescence was continuously measured. The luminescence signal was evaluated after it reached plateau, and the average of 13 luminescence data points (representing a 6-min period) was evaluated. In each experiment, data were normalized to the luminescence value after vehicle pretreatment. Thus, 100% reflects the maximum cAMP signal.

*Bioluminescence resonance energy transfer measurements*. HEK 293T cells were transfected in solution with HA-$D_3R$, $Gα_{i1}$–Rluc8, Venus–$β_1$ or $β_1$, and $γ_2$ plasmid DNA using the calcium phosphate precipitation method. The next day we replaced

the medium with modified Kreb's Ringer medium and performed BRET experiments using a Thermo Scientific Varioskan Flash multimode plate reader similarly as previously described[78]. After the addition of the luciferase substrate coelenterazine $h$ (5 μM), we measured luminescence intensities at 530 nm and 480 nm using filters to determine the BRET ratio (luminescence$_{530nm}$/luminescence$_{480nm}$). Baseline BRET ratio was measured for four cycles, two additional cycles were performed in the case of pretreatment with vehicle or fluo-cariprazine, then stimuli were applied and luminesce intensities were detected for 20 cycles. To calculate the BRET response, we first subtracted the BRET ratio of cells coexpressing the untagged $\beta_1$ subunit from the BRET ratio of cells coexpressing Venus-$\beta_1$. Then we corrected the baseline BRET ratio and subtracted the BRET ratio of vehicle-treated cells from the average BRET ratio of stimulus-treated cells. All BRET measurements were performed in triplicates.

*2-Arachidonoylglycerol-based fluorescence assay for DH-463.* The natural substrate assays were performed as reported previously[80,81]. All chemicals and reagents were purchased from Sigma-Aldrich, unless mentioned otherwise. Standard assay conditions: 0.2 U/mL glycerol kinase (from Cellulomonas sp. Sigma, G6142), glycerol-3-phosphate oxidase (from *Streptococcus thermophilus*, Sigma, G4388) and horseradish peroxidase (from Horseradish, Sigma, 77882), 0.125 mM ATP, 10 μM Amplifu™Red, 5% DMSO in a total volume of 200 μL. 2-Arachidonoylglycerol was from Cayman Chemical. All measurements were performed in $n = 4$ or $n = 8$ for controls, with Z' ≥ 0.6.

**Activity-based protein profiling.** Activity-based protein profiling was performed as reported previously[82]. Briefly, rodent brain membrane proteome (2 mg/mL, 19 μL) was incubated with DMSO or inhibitor in 0.5 μL DMSO for 30 min at room temperature. For competitive activity-based protein profiling, JZL184 preincubation was followed by 80 nM DH-463 incubation for 30 min at room temperature, before the reaction was quenched with 10 μL standard 3 × Laemmli sample buffer. The samples were loaded and resolved on SDS PAGE gel (10% acrylamide). The gels were scanned using a ChemiDoc MP system and analyzed using Image Lab 4.1.

**Preparation and labeling of primary neuronal cultures.** Hippocampi were dissected from C57BL/6 mouse pups at postnatal stages P0/P1, and collected in dissection solution (116 mM NaCl, 5.4 mM KCl, 26 mM NaHCO$_3$, 1.3 mM NaH$_2$PO$_4$, 2 mM MgSO$_4$, 2 mM CaCl$_2$, 0.5 mM EDTA, 25 mM D-glucose). A 15-min-long papain treatment (15 U/ml, Sigma) at 37 °C was followed by trituration with fire-polished Pasteur pipette in dissection solution with 1% BSA. Cells were collected by centrifugation (200×$g$ for 3 min, at room temperature), and resuspended in Neurobasal A maintenance medium (Gibco) supplemented with L-glutamine (2 mM final concentration, Sigma) and B-27 (20 ml/l (2%), Gibco)). Viable cell density was estimated with Trypan blue (Sigma) in a hemocytometer, and cells were plated in maintenance medium supplemented with 10% NHS (Normal Horse Serum, Sigma) and gentamycin (10 μg/ml, Sanofi-Aventis) with ~50 cells/mm$^2$ density on cleaned 18-mm poly-D-lysine-coated coverslips in a 12-well dish. After an hour, the plating medium was gently exchanged with maintenance medium. Half of the total amount of media was changed every 3 days. At DIV7-15, neurons were transiently transfected with HA-D$_3$R plasmid using Lipofectamine 2000. The labeling procedures and microscopic settings were almost identical to those of HA-D$_3$R expressing HEK 293 cells. After incubation with fluo-cariprazine and anti-HA primary and secondary antibodies, the neurons were fixed with 4% PFA for 10 min, washed three times with PB for 10 min, and then were treated with 0.1% Triton, 5% NDS, and 1% BSA (15 min) for simultaneous permeabilization and blocking before incubation with anti-microtubule-associated protein 2 (MAP2) antibody. The primary antibody was diluted in PB containing 0.5% NDS and 0.1% BSA, incubated for 30 min, and stained with Alexa488-anti-rabbit secondary antibody for 15 min. The samples were further processed for correlated confocal and PharmacoSTORM imaging.

**Fluorescence intensity measurement of single pharmacoprobes/antibodies.** Fluo-cariprazine and secondary antibodies were diluted in filtered DPBS (1 nM and ~0.5 ng/ml, respectively) and applied to #1.5 coverslips. Control coverslips only held filtered DPBS. Samples were imaged via a CFI Apo TIRF ×100 Oil 1.49 NA objective on a Ti-E inverted microscope equipped with an N-STORM system, a Nikon C2 confocal scan head, and an Andor iXon Ultra 897 EMCCD camera operated by NIS-Elements AR software 4.51 (Nikon). We applied 1× beam-focusing lens in the Nikon STORM-TIRF illuminator and images were acquired by using low intensity 647 nm laser power with the N-STORM filter cube (EX:DM:390-650, DM:660 nm, EM:670–760 nm). We recorded 500 frames (exp.:50 ms, 160 nm/pixel, 256 × 256 pixels). All analyses steps were performed in NIS-Elements AR 5.21.01 (Nikon) software. Background subtraction was done using the rolling ball algorithm, centroid pixels of the fluorescent events were identified using the bright spot detection algorithm, and binaries were dilated by the factor of 2 which resulted in the region of interests (ROIs) with the shape of 5 × 5 pixels encompassing the PSF of fluorescent events. Large ROIs indicating events that are too close to be separated were discarded. Intensity measurements were done in ROIs over time and the remaining backgrounds were subtracted.

Stepwise fading of the fluorescent intensity traces of the secondary antibodies could be readily observed by visual inspection. After complete fading of the signal (bleaching), the fluorescent event occasionally reappeared for a short time due to the reactivation of a fluorophore. By analyzing these step-like events, the fluorescent intensity of a single fluorophore could be estimated. Fluo-cariprazine molecules were only shortly adhered to the coverslips, therefore their bleaching, potential stepwise fading, or separation from the coverglass could not be reliably discerned.

**Animals.** All experiments were approved by the Hungarian Committee of the Scientific Ethics of Animal Research (license number: 2018/1 internal license number and PE_EA_49-5_2020) and were carried out according to the Hungarian Act of Animal Care and Experimentation (1998, XXVIII, Section 243/1998), in accordance with the European Communities Council Directive of November 24, 1986 (86-609-EEC; Section 243/1998). Mice were kept under approved, specific-pathogen-free laboratory conditions (12-h light/12-h dark cycle, 22–24 °C, 40–70% humidity), and all efforts were made to minimize pain and to reduce the number of animals used. C57BL/6 mice were obtained from Charles River Laboratories. The mouse line bearing a targeted mutation in the *Drd3* gene has been validated in earlier studies[83]. Male and female mice (25–57 days old) were used in the study.

**Acute slice preparation.** Mice were decapitated under deep isoflurane anesthesia, the brains were carefully removed from the skull and rapidly transferred into ice-cold sucrose-containing artificial cerebrospinal fluid (sucrose-ACSF; containing in mM: 75 NaCl, 75 sucrose, 2.5 KCl, 25 glucose, 1.25 NaH$_2$PO$_4$, 4 MgCl$_2$, 0.5 CaCl$_2$, and 24 NaHCO$_3$, Sigma-Aldrich, St. Louis, MO, USA), constantly equilibrated with 95% O$_2$ and 5% CO$_2$. Acute coronal slices (anteroposterior 1.7 mm to 0.2 mm from bregma) at 300 μm thickness were cut with a Leica VT-1200S Vibratome (Nussloch, Germany).

**Electrophysiological experiments.** Coronal slices that were subjected to electrophysiological measurements were first incubated in sucrose-ACSF for 1 h at 34 °C. Afterward, the slices were kept in the oxygenated incubation chamber at room temperature until recordings. Whole-cell patch-clamp recordings were made in a submerged recording chamber at 33 °C constantly perfused with oxygenated ACSF solution (in mM: 126 NaCl, 2.5 KCl, 10 glucose, 1.25 NaH$_2$PO$_4$, 2 MgCl$_2$, 2 CaCl$_2$, and 26 NaHCO$_3$, Sigma-Aldrich, St. Louis, MO, USA). Slices were visualized with an upright Nikon Eclipse FN1 microscope equipped with infrared differential interference contrast (DIC) optics (Nikon, Tokyo, Japan). Current-clamp recordings were obtained from granule cells of the Islands of Calleja with borosilicate glass pipettes (0.86 mm inner diameter and 1.5 mm outer diameter with 8–10 MΩ resistance) filled with internal solution (containing in mM: 126 K-gluconate, 4 KCl, 10 HEPES, 0.2 EGTA, 4 Mg-ATP, 0.3 Na$_2$-GTP, 10 phosphocreatine and 8 biocytin, Sigma-Aldrich, St. Louis, MO, USA; pH 7.2; 290 mOsm). Pipette capacitance neutralization (2.5–5 pF remaining capacitance) and bridge balance compensation were set to eliminate apparent voltage offsets upon current steps. Recordings were performed using MultiClamp 700B amplifier (Molecular Devices, San José, CA, USA). Signals were filtered at 3 kHz using a Bessel filter and digitized at 20 kHz with Digidata 1440 A analog-to-digital interface (Molecular Devices). The recorded traces were analyzed using the Clampfit 10 software (Molecular Devices). Granule cell resting membrane potentials and firing patterns were studied in current-clamp mode, using square current steps of 500 ms duration, which started from −10 pA with 1 pA increments. Then, cells were clamped to −65 mV for 5–10 min to allow whole-cell labeling with biocytin. After dye filling, the slices were treated with 300 nM fluo-cariprazine (described in the next paragraph).

**Pharmacoprobe treatment of acute slices.** Acute slices from 26–33-days-old *Drd3*$^{+/+}$ and *Drd3*$^{-/-}$ littermates were transferred into oxygenated ACSF containing 300 nM fluo-cariprazine and incubated for 40 min at room temperature. In the case of competitive ligand-binding experiments, slices were pretreated with 30 μM cariprazine or DMSO (vehicle) for 10 min, then 300 nM fluo-cariprazine was added to the incubation chambers for an additional 30 min. Sections were rinsed three times in ACSF and then submerged into ice-cold 4% paraformaldehyde (PFA) in 0.1 M PB and fixed overnight at 4 °C. Acute slices containing biocytin-filled granule cells were stained with fluo-cariprazine in the same manner. After ligand treatment, slices were kept at 4 °C and processed for different post hoc imaging experiments, as described below. All samples, which were ready for imaging, were dried onto coverslips or glass slides. Microscopic images were always acquired within 1 or 2 days of ligand treatment to minimize potential ligand dissociation. To measure background STORM signal in tissue, control sections were processed in the same way except that they were not treated with fluorescent ligands.

**Three-dimensional analysis of fluo-cariprazine binding.** Mice were anesthetized with intraperitoneal injection of avertin (1.25% v/v, Sigma), then transcardially perfused with oxygenated, ice-cold ACSF for 3 min. The brain was carefully removed from the skull and placed into ice-cold sucrose-containing artificial

cerebrospinal fluid. The 2.5-mm-thick acute slices were cut with a Leica VT-1200S Vibratome. Slices were incubated in oxygenated ACSF with 300 nM fluo-cariprazine for one hour, rinsed three times with ACSF, and fixed overnight with 4% PFA at 4 °C. Free-floating slices at 40-μm thickness were cut, fixed with 4% PFA for 10 min, washed three times in PB for 5 min. Thereafter, immunolabeling and washing steps were performed as described later in "Immunolabeling on fluo-pharmacoprobe-treated tissue sections". DAPI (1:2000) was added to the sample in conjunction with secondary antibodies. Slices were mounted on glass slides in ProLong™ Diamond Antifade Mountant. Fluorescent images were taken with 3DHISTECH Panoramic MIDI II slide scanner, using a Zeiss Plan-Apochromat ×20 objective (0.8 NA). Consecutive sections were aligned with a built-in plugin for feature-based image registration in the NIS-Elements AR 5.21.01 (Nikon) program. The hilar regions of the Islands of Calleja were manually demarcated based on DAPI and fluo-cariprazine labeling and converted into binary layers. 3D meshes were extracted from binary stacks with 3D Slicer 4.10.2 software[84]. The neuroanatomical map of fluo-cariprazine binding into 3D mouse brain reference space was fitted into the brain map obtained from the 3D Brain Atlas Reconstructor[85]. All structures were visualized with Blender 2.80.

**Morphological analysis of in vitro-recorded cells.** The following steps were executed at 4 °C, and all reagents were dissolved in pre-chilled 0.1 M PB. Fixed sections were washed three times with PB for 10 min then treated with 0.5% Triton X-100 for 20 min and incubated in Alexa Fluor 488-Streptavidin (1:1000) for 1 h with or without DAPI (1:2000). After PB wash (three times 10 min), the slices were mounted on glass slides and immediately transferred to a Nikon A1R confocal laser-scanning system built on a Ti-E inverted microscope operated by NIS-Elements AR software 4.50.00 (Nikon). Those slices that were further used for PharmacoSTORM experiments were mounted in PB, otherwise, the mounting medium was ProLong™ Diamond Antifade Mountant. Low-magnification image stacks were collected using a CFI Plan Apo VC ×20 Air 0.75NA objective. After confocal imaging, the slices were directly embedded in 2% agarose, and 20-μm-thick coronal sections were cut with a Leica VT-1000S Vibratome in ice-cold PB. The sections were fixed with 4% PFA for 10 min, washed in PB three times for 5 min, and mounted and dried on acetone-cleaned #1.5 borosilicate coverslips. The complete protocol, including the subsequent PharmacoSTORM imaging, was performed on the same day. Representative examples of biocytin-filled neurons were reconstructed from confocal image stacks using Neurolucida® software, or NIS-Elements AR 5.21.01 (Nikon). 3D meshes were created and visualized as described in the previous section, using 3D Slicer 4.10.2 and Blender 2.80 software.

**Immunolabeling on fluo-pharmacoprobe-treated tissue sections.** All following procedures were carried out at 4 °C, in pre-chilled buffers. Fluo-cariprazine stained and fixed 300-μm-thick brain slices were re-sectioned with a Leica VT-1200S Vibratome. For low-resolution fluorescence microscopy, 40-μm thin sections were cut, whereas 20-μm thin sections were used for STORM imaging. All slices were fixed with 4% PFA for 10 min and washed with PB three times for 5 min before starting a free-floating immunolabeling protocol. Samples were treated with 5% NDS, 1% BSA, 0.3% Triton in TBS (Tris-buffered saline) for 45 min for blocking and permeabilization, and immunolabeled overnight with primary antibodies (details listed in Supplementary Table S2). Following three 10-m washing steps in TBS, the corresponding fluorescent secondary antibodies (details listed in Supplementary Table S2) were incubated for 2 h. For confocal microscopy, slices were mounted on glass slides in ProLong™ Diamond Antifade Mountant and imaged with a Nikon A1R confocal laser-scanning microscope.

**In vivo competitive drug-binding measurement.** Male 26–33-days-old C57BL/6 mice were intraperitoneally injected with 1 mg/kg cariprazine or an equal volume of vehicle (0.9% saline + 1% Tween80). After injection (2 h), pharmacological tissue labeling was carried out similarly to the previous experiments. Acute coronal brain slices were prepared as previously described, and slices were incubated with 30 nM fluo-cariprazine in oxygenated ACSF for 10 min. Slices were rinsed three times with ACSF and fixed overnight with 4% PFA. Slices were either processed for correlated confocal and STORM imaging or for conventional confocal imaging. In the latter case, 40-μm thin slices were cut, stained with DAPI (1:2000, 15 min), washed with PB three times for 10 min, mounted in ProLong™ Diamond Antifade Mountant, and imaged with Nikon A1R confocal laser-scanning microscope.

**Fluo-pharmacoprobe treatment of fresh-frozen brain cryosections.** The brains of Drd3[+/+] and Drd3[−/−] littermates were removed and frozen on isopentane chilled with dry ice. The brain tissue was equilibrated in the cryostat for 2 h at −20 °C, then 20-μm-thick cryosections were collected on poly-D-lysine-coated coverslips. All following steps were executed at 4 °C. Fluo-cariprazine treatment in TBS for 40 min was followed by three 5-m long washing steps, then sections were fixed with 4% PFA, and fixative was washed out by TBS (three times 10 min). For confocal imaging, nuclear staining was performed with DAPI (1:1000, 15 min), and slices were mounted on glass slides in ProLong™ Diamond Antifade Mountant and imaged with Nikon A1R confocal laser-scanning microscope. For Pharmaco-STORM imaging, details are described in the following section.

**Electron microscopical investigation of individually labeled neurons.** Acute slice-electrophysiology and biocytin labeling of individual neurons were performed in the same manner as described earlier. After 15–20 min of recording, cells were left in the chamber for few m to allow fine diffusion of biocytin, then, slices were immersion-fixed (4% PFA and 2% glutaraldehyde (GA, Sigma-Aldrich, St. Louis, MO, USA) in 0.1 M PB). Following a 24 h-fixation period, 300-μm-thick slices were extensively washed in 0.1 M PB, then samples were re-sectioned to 100-μm thin sections. After slicing and washing in 0.1 M PB, the sections were incubated in 10% sucrose for 15 min and 30% sucrose overnight, followed by freeze-thawing over liquid nitrogen four times. Sections were then treated with 1% H₂O₂ for 10 min and washed twice for 5 min with 0.1 M PB and 0.05 M TBS, respectively. All washing steps and dilutions of the avidin-biotin complex (ABC, Thermo Scientific) were performed in 0.05 M TBS, pH 7.4. Sections were incubated in ABC solution for 90 min, followed by subsequent washing in TBS and TB. Tissue bound ABC was visualized with 3,3-diaminobenzidine (DAB) chromogen (ImmPACT® DAB Substrate, Peroxidase (HRP) SK-4105, Vector Laboratories) in 0.05 M TB for 15 min. After development, the sections were treated with osmium tetroxide (0.5%) in PB for 20 min at 4 °C in the dark and dehydrated in an ascending series of ethanol and acetonitrile, before being embedded in Durcupan (ACM, Fluka, Buchs, Switzerland). During dehydration, sections were treated with 1% uranyl acetate in 70% ethanol for 15 min at 4 °C in the dark. Area of interest containing the labeled cell in the Islands of Calleja region was re-embedded and re-sectioned to produce 60-nm ultrathin sections with a Leica EM UC6 Ultramicrotome (Leica Microsystems). Ultrathin serial sections were collected on Formvar-coated single-slot grids and further stained with lead citrate. Electron microscopical images were taken at various magnifications (×5000–30,000) with a Hitachi 7100 electron microscope.

**Correlated confocal and PharmacoSTORM imaging and dual-color direct STORM imaging.** For STORM imaging, the HEK 293 cells and the brain sections were covered in freshly prepared STORM imaging medium (containing 5% glucose, 0.1 M mercaptoethylamine, 1 mg/ml glucose oxidase, and 1500 U/ml catalase in DPBS), and sealed with nail-polish ~10 min before image acquisition. STORM super-resolution images and the correlated high-power confocal stacks were acquired via a CFI Apo TIRF ×100 Oil 1.49 NA objective on a Ti-E inverted microscope equipped with an N-STORM system, a Nikon C2 confocal scan head, and an Andor iXon Ultra 897 EMCCD camera. Additional details of correlated confocal and STORM image acquisition have been described in the previous studies[8,30].

In cell culture experiments, field-of-view selection of clear cross-sections of cell membranes was based on the live EMCCD camera image. The focal plane was further optimized by confocal z-stacks of the plasma membrane of HEK 293 cells or primary neuronal processes. In brain section experiments, the hilus of the Islands of Calleja was in focus in low-magnification live images based on fluo-cariprazine labeling, or neuronal nitric oxide synthase (nNOS)-immunopositivity. The best imaging plane of tyrosine hydroxylase-positive structures and biocytin-filled granule cell processes were subsequently assessed after high-power confocal z-stacks. Measurements were performed at ~5 μm depth in the sections. The anatomical borders of the ventral and dorsal striatal areas were defined by DARPP-32 immunolabeling.

For all PharmacoSTORM probes, we performed direct STORM (dSTORM) imaging in continuous activation mode using 100% 647 nm imaging and 10% 405-nm activation laser power with a far-red STORM filter cube (EX:- DM: 660 nm, EM: 670–760 nm). ImmunoSTORM targets were visualized by two-step immunohistochemistry using CF568-conjugated secondary antibodies, and images were acquired after PharmacoSTORM image acquisition in a sequential manner with a HQ Red filter cube (EX: 554–568 nm, DM: 575 nm, EM: 557–661 nm). During cell culture experiments, we applied ×2 beam-focusing lens in the Nikon STORM-TIRF illuminator, used the astigmatism lens for 3D, and captured 5000 frames per image. In tissue preparations, we used a ×4 beam-focusing lens and collected 10,000 frames per image. The focus was stabilized by the Nikon PFS hardware-based focus-lock.

**PharmacoSTORM image analysis.** PharmacoSTORM and ImmunoSTORM images were processed and visualized with the N-STORM module in NIS-Elements AR software. The threshold settings were optimized for each PharmacoSTORM probe and fluorescent secondary antibody. At each analysis assay, identical analysis parameters were used for all images.

To analyze the number of localization points (LPs) in membranes of cultured cells, we merged the dSTORM images obtained from pharmacolabeling (PharmacoSTORM) and immunolabeling (ImmunoSTORM). The plasma membrane of HEK 293 cells was contoured in the VividSTORM software to determine the ROI for further analysis[30]. Custom-written Python scripts were used to count localization points in manually delineated plasma membrane ROIs. To avoid any quantification problems that could arise from the non-homogenous illumination caused by the STORM-TIRF magnification lens, only the data from the inner 64 × 64 pixels on the EMCCD camera (covering a total of 104.86 μm²) were analyzed in case of cell culture samples and data from 32 × 32 pixels on the EMCCD camera (covering a total of 26.21 μm²) were analyzed in case of correlated confocal and STORM imaging in tissue samples. When necessary, the confocal

image was translated manually to fit the image of the same structure acquired on the EMCCD sensor in epifluorescence illumination.

To demonstrate the intensity of fluo-cariprazine binding in different regions, the number of STORM LPs was determined in the inner quarter of the field-of-view. Density values were normalized to STORM background (measured on control slices without fluo-cariprazine treatment), then the background was subtracted. To compare the overall density of STORM LPs in the brain under different experimental conditions, several STORM images were taken in the hilar regions of the Islands of Calleja, the number of LPs was determined in the inner sixteenth of the field-of-view. To characterize how the nanoscale fluo-cariprazine distribution is related to other brain structures, we applied a similar workflow as described in our prior publication[8,30]. The confocal stacks of the biocytin-filled neuronal processes or the tyrosine hydroxylase-positive varicosities were deconvolved with 100 iterations of the Classic Maximum Likelihood Estimation algorithm in Huygens software (SVI, The Netherlands). The maximum intensity z-projection of three image planes centered at the plane corresponding to the focal plane of the STORM image was used for further analysis. The original 80 nm/pixel confocal image was resampled to 10 nm/pixel, and the LPs from the STORM data were overlaid on it. The confocal image was converted to a binary image based on a threshold value, defined in NIS-Elements AR software, and the density of overlapping fluo-cariprazine STORM LPs was quantified. We compared fluo-cariprazine LP density over the investigated structure to the average LP density in the adjacent area within the same image by repeatedly dilating the binary areas 50 times. Thereafter, we evaluated the ratio of fluo-cariprazine LP density on the actual biological structure (original binary) to the average density on the dilated area. If this ratio equals 1 then a structure holds an average density of LPs, >1 if it is a fluo-cariprazine-rich nanoscale profile, and <1 if fluo-cariprazine molecules avoid the respective profile. To validate our analysis, we performed the same measurement after the distribution of the fluo-cariprazine LPs was randomized. In this case, the binary areas on the actual biological structure are always expected to hold the average density of LPs.

To compare fluo-cariprazine distribution at different axonal segments, the subcompartments were classified by visual inspection either into the terminal or preterminal segments (enlarged varicosities or thinner segments connecting the varicosities, respectively) in resized confocal images. ROIs were defined by their contour's coordinates and the density (number of LPs in ROI area) were calculated. To determine whether there is an accumulation of neighboring fluo-cariprazine LPs at given nanoscale distances from TH-positive boutons, a 2D convex hull was fitted onto the TH ImmunoSTORM LPs, and the distance of each fluo-cariprazine LP from the nearest convex hull surface was calculated. A steady increase of the cumulative distribution curve of fluo-cariprazine distances indicates the independent distribution of fluo-cariprazine LPs from dopaminergic structures, whereas accumulation/depletion of fluo-cariprazine molecules at a certain distance from the convex hull surface would result in a leftward/rightward shift in the cumulative distribution curve. Only fluo-cariprazine LPs that are ≤1000 nm distant from the convex hull surface were analyzed. For image representation or analyses, ROIs containing TH ImmunoSTORM LPs from dopaminergic boutons were first selected and contoured in the VividSTORM software as previously described[8,30]. 2D polygons were fitted by custom-written Python scripts. Boutons were only included in the analysis if a convex hull could be adequately fitted.

STORM data on biocytin-filled granule cells were visualized by Visual Molecular Dynamics (VMD) 1.9.3 software[86]. The 3D localization coordinates were converted to PDB (Protein Data Bank) file format, and STORM data were assigned to solid Van der Waals spheres for atoms.

**Statistics and reproducibility.** The sample size ("$n$") in Figure legends refers to the number of independent experiments (biological replicates). For cell culture experiments, "$n$" represents independent transfection and labeling procedures, where multiple cells from multiple wells were evaluated in each experiment. For tissue experiments, "$n$" represents the number of animals, and each data point represents the average density of STORM localization points taken throughout the entire Islands of Calleja brain region from each animal. For the analysis of correlated PharmacoSTORM and confocal images, "$n$" represents the number of images that were taken from the brains of the indicated number of animals. If the results of two different experimental conditions (e.g., the effect of a vehicle vs. antagonist pretreatment on pharmacoprobe binding) are displayed side-by-side, images were always selected from a single experiment, where different treatment groups were handled in parallel. All microscopic images are representative images of at least three biologically independent experiments. Confocal images of cell cultures in Fig. 1a(III), and 2d are representative of three independent transfections and labeling procedures. Microscopic images in Figs. 3a, b, 3f, g, 4a, b, 4e, and 5b are representative of experiments from three different animals. All images of granule cells in Figs. 6a, 6c–h, 7a, 7c–f, and 7h show observations that were made in at least three different animals. The same samples were never re-measured. GraphPad Prism software was used for graph construction, statistical comparisons, and curve fitting. Unless otherwise stated, data are presented as mean ± SEM. The statistical tests used in each experiment are described in their respective Figure legend.

**Reporting summary**. Further information on research design is available in the Nature Research Reporting Summary linked to this article.

## Data availability

All materials and data are available from the corresponding authors upon reasonable request. DAPI and fluo-cariprazine channels of multi-channel microscopic images are shown separately in the source data file. Source data are provided with this paper.

## Code availability

Python-based scripts that were written for easier and rapid handling of data files and to analyze STORM images as indicated in "Methods", are available from the authors upon request.

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

## Acknowledgements

The authors are grateful to B. Pintér and E. Tischler for their laboratory support. The authors thank Drs. J. Brunner, K. Mackie, P. Somogyi, and J. Szabadics for valuable discussions and comments on the manuscript. The help of Drs. P. Vági, C. Pongor, the Nikon Microscopy Center at the Institute of Experimental Medicine, Nikon Europe B.V., Nikon Austria GmbH, and Auro-Science Consulting is acknowledged for kindly providing microscopy support. The authors are also grateful to Drs. E. Horváth, K. Kenesei, M. Kisfali, Z. Lele, Z. László, G. Balla, F. Mógor, and J. Glavinics for their experimental advice in laboratory work. The authors are indebted to Dr. B. Gereben for providing the Grenier Luminometer and HEK 293 cells. The authors also thank the help of D. Kiss and Á. Kelemen for their help in chemical modeling and synthesis, and M. Jiang for performing the docking studies with DH-463. This study was supported by the National Brain Research Program (2017-1.2.1-NKP-2017-00002 for I.K. and G.M.K.), by the National Research, Development and Innovation Office, Hungary (VEKOP-2.3.3-15-2016-00013 for super-resolution microscopy development for I.K.; Frontier Program 129961 for cannabinoid research for I.K.; postdoctoral fellowship PD124598 for P.Á.-B., EFOP-3.6.3-VEKOP-16-2017-00009 scholarship for M.V.). I.K. holds the Naus Family Chair in Addiction Sciences in the Department of Psychological and Brain Sciences at Indiana University Bloomington and his work is also supported by the National Institutes of Health (R01NS099457 and R01DA044925). This study was also supported by the New National Excellence Program of the Ministry for Innovation and Technology (ÚNKP-20-3-II-SE-33 and ÚNKP-19-3-III-SE-16 to S.P. and B.B.); by the VICI-grant from the Netherlands Organization for Scientific Research and funding from Oncode Institute to M.v.d.S.; the Italian Ministry of University and Research (PRIN 2017-Prot. 201779W93T to G.M.L.) and the University of Catania Intramural Funds (Starting Grant 2020 to G.M.L.).

## Author contributions

S. Prokop designed and performed microscopic imaging, analyzed the data, prepared the figures, and wrote the manuscript. P. Ábrányi-Balogh designed and synthesized fluorescent drugs, prepared the corresponding figures, and wrote the manuscript. B. Barti performed patch-clamp recordings, electron microscopic investigations, and analyzed the corresponding data. M. Vámosi analyzed images for 3D reconstruction. L. Barna assisted and supervised all microscopic experiments and image analysis. M. Zöldi performed STORM data analysis. G.M. Urbán directed and performed combined pharmaco-and immunolabeling. A.D. Tóth designed and performed BRET measurements. B. Dudok performed super-resolution microscopy. A. Egyed participated in the design and synthesis of fluo-cariprazine. H. Deng synthesized and characterized DH-463. G.M. Leggio provided *Drd3* knockout mouse line. L. Hunyady supervised BRET measurements. M.v.d. Stelt supervised the synthesis of DH-463. G.M. Keserű and I. Katona conceived and supervised the project, wrote the manuscript and their work made equal contribution to the study. All authors commented on the manuscript.

## Competing interests

The authors declare no competing interests.
