## [Peer Review File · Nature Communications]

Pharmacostorm nanoscale pharmacology reveals cariprazine binding on Islands of Calleja granule cellsEditorial Note: This manuscript has been previously reviewed at another journal that is not operating a transparent peer review scheme. This document only contains reviewer comments and rebuttal letters for versions considered at *Nature Communications*.

REVIEWER COMMENTS

Reviewer #3 (Remarks to the Author):

This is a revision of a manuscript previously reviewed in another journal. I appreciate the authors' efforts to address issues raised in the previous round of the review. PharmacoSTORM can become a powerful strategy to examine nano-scale localization of various receptors in tissues. As I wrote previously, a less satisfying aspect of this work is that the unique applications that are only possible with this strategy are not effectively demonstrated. For example, Figures 4 and 5 would be possible even with confocal microscopy. However, these applications should come in the future and I agree that it is more constructive to publish this work early for broader usage in the research community. I have no further requests.

Reviewer #4 (Remarks to the Author):

In the present manuscript the author develops a suite of pharmacological tools to investigate the localization of protein targets at nanoscale resolution. They introduce fluorescently labelled small molecules that bind to 3 different types of molecules (GPCR, enzyme and ion channel) for dSTORM imaging and characterize in more detail fluo-cariprazine, a fluorescent derivative of a clinically approved drug for which the mechanism of action is poorly characterized. Using this tool, they show specificity for D3R and localize binding site for the drug at macro and nano scale. The authors identify the Islands of Calleja as a prominent region for fluo-cariprazine binding, characterize granule cells in this region, and show at a subcellular level that 1 – binding sites are excluded from axons of dopaminergic neurons 2 – binding sites are homogeneously distributed along axons of granule cells.

The body of work is compelling and the manuscript well written with an extended method section that provides enough information to replicate or expand the presented results. I envision the presented tools and pharmacoSTORM method to have a significant impact in the neuroscience and microscopy communities as, as the authors claim, there has been a lasting lack of tools for localizing certain proteins of interest. While I thoroughly enjoyed the manuscript, I have remaining concerns around the performance achieved by the pharmacoSTORM approach that I would like addressed by the authors.

Major comment:

While fluorescently labelled small molecules have been used for years to localize receptors, their use for single molecule localization microscopy is new to my knowledge. However, such methods come at a cost: any non-specific labelling will be impossible to distinguish from the specific labelling once investigated at a single molecule level where signal and noise are confounded. This becomes particularly challenging when investigating low abundance proteins such as GPCRs (low signal, point 1), or when using probes that have significant non-specific interactions with the sample (high noise, point 2). These limitations are inherent to the approach, and while they do not contradict the claims made by the authors in the present study, they should be addressed or discussed to ensure a proper use of the method.

1- While the Islands of Calleja is a region with dramatic enrichment in fluorescent signal from fluo-cariprazine, it is unlikely to be the only region where binding sites are present. This is reflected in the present work with the low but existent binding to D2R versus D3R, D3R-KO data that still show fluo-CAR detection; but also from the literature that indicates that D3R is expressed in other brain regions. From the results presented in the current study, it is not clear what the performance of pharmacoSTORM is in brain regions where the expression of true binding sites is lower.

2- Cell models, competition assays as well as the D3RKO data are powerful but do not encompass the full repertoire of non-specific binding sites that can be present in a complex biological sample. Interaction of the fluorescent probe with the biological material (in tissue particularly) would result in artifactual localization point. Such contamination of the signal is inherently challenging to discriminate from true binding sites in single molecule localization microscopy. Non-specific binding

would seem particularly challenging to overcome with highly hydrophobic probes such as cannabinoid-based compounds. In the case of non-covalent binders, it is unclear to me whether the off-rate of compounds is sufficient to wash out unbound probes while the staining on real binding sites is preserved.

In consideration of these two points, I suggest one additional experiment to the authors. The authors would provide pharmacoSORM images of brain structures with lower D3R expression levels and repeat the nearest neighbor analysis as in Fig 5k. This would confirm that 1) single molecule detection can detect functionally relevant binding sites at low density 2) Signal to noise is sufficient to discriminate binding sites from non-specific binding.

This experiment would, in my opinion, validate the power of fluo-CAR for pharmacoSORM. Additionally, I would appreciate that these limitations are discussed in regard to the generalization of the pharmacoSORM approach, as well as more details for the washing steps (duration) in the material and method section.

Minor comment:

- The reader cannot estimate the degree of labelling in other brain regions based on the images in figure 3. Ideally, the authors would provide images without the DAPI merge (provides little information to the reader and impedes the reading of the images both a low and super resolution level) as well as images with dramatically increased contrast (saturating the signal in the Islands of Calleja) for panels 3a,b,f,g; 5b.

Damien Jullié, PhD.

Reviewer #3:

- This is a revision of a manuscript previously reviewed in another journal. I appreciate the authors' efforts to address issues raised in the previous round of the review. PharmacoSTORM can become a powerful strategy to examine nano-scale localization of various receptors in tissues. As I wrote previously, a less satisfying aspect of this work is that the unique applications that are only possible with this strategy are not effectively demonstrated. For example, Figures 4 and 5 would be possible even with confocal microscopy. However, these applications should come in the future and I agree that it is more constructive to publish this work early for broader usage in the research community. I have no further requests.

We would like to thank the Reviewer for his/her acknowledgement that "PharmacoSTORM can become a powerful strategy". We also thank his/her stimulating opinion that a broad usage of PharmacoSTORM in the research community is expected. We do hope that the more than 30 quantitative datasets derived from nanoscale molecular localization information collected from single cells and tissue preparations in this study will facilitate the interest for the PharmacoSTORM approach in the life science community that will culminate in the development of further applications by our colleagues.

Reviewer #4:

- The body of work is compelling and the manuscript well written with an extended method section that provides enough information to replicate or expand the presented results. I envision the presented tools and pharmacoSTORM method to have a significant impact in the neuroscience and microscopy communities as, as the authors claim, there has been a lasting lack of tools for localizing certain proteins of interest. While I thoroughly enjoyed the manuscript, I have remaining concerns around the performance achieved by the pharmacoSTORM approach that I would like addressed by the authors.

We appreciate very much that the Reviewer took his/her time to carefully read our manuscript and to provide his/her stimulating opinion and constructive suggestions. In addition, we are delighted to learn that the Reviewer shares our view that PharmacoSTORM will "have a significant impact in the neuroscience and microscopy communities". We are also glad that the Reviewer "thoroughly enjoyed the manuscript."

- Major comment:
While fluorescently labelled small molecules have been used for years to localize receptors, their use for single molecule localization microscopy is new to my knowledge. However, such methods come at a cost: any non-specific labelling will be impossible to distinguish from the specific labelling once investigated at a single molecule level where signal and noise are confounded. This becomes particularly challenging when investigating low abundance proteins such as GPCRs (low signal, point 1), or when using probes that have significant non-specific interactions with the sample (high noise, point 2). These limitations are inherent to the approach, and while they do not contradict the claims made by the authors in the present study, they should be addressed or discussed to ensure a proper use of the method.

1- While the Islands of Calleja is a region with dramatic enrichment in fluorescent signal from fluo-cariprazine, it is unlikely to be the only region where binding sites are present. This is reflected in the present work with the low but existent binding to D2R versus D3R, D3R-KO data that still show fluo-CAR detection; but also from the literature that indicates that D3R is expressed in other brain regions. From the results presented in the current study, it is not clear what the performance of pharmacOSTORM is in brain regions where the expression of true binding sites is lower.

2- Cell models, competition assays as well as the D3RKO data are powerful but do not encompass the full repertoire of non-specific binding sites that can be present in a complex biological sample. Interaction of the fluorescent probe with the biological material (in tissue particularly) would result in artifactual localization point. Such contamination of the signal is inherently challenging to discriminate from true binding sites in single molecule localization microscopy. Non-specific binding would seem particularly challenging to overcome with highly hydrophobic probes such as cannabinoid-based compounds. In the case of non-covalent binders, it is unclear to me whether the off-rate of compounds is sufficient to wash out unbound probes while the staining on real binding sites is preserved.

In consideration of these two points, I suggest one additional experiment to the authors. The authors would provide pharmacOSTORM images of brain structures with lower D3R expression levels and repeat the nearest neighbor analysis as in Fig 5k. This would confirm that 1) single molecule detection can detect functionally relevant binding sites at low density 2) Signal to noise is sufficient to discriminate binding sites from non-specific binding.

This experiment would, in my opinion, validate the power of fluo-CAR for pharmacOSTORM. Additionally, I would appreciate that these limitations are discussed in regard to the generalization of the pharmacOSTORM approach, as well as more details for the washing steps (duration) in the material and method section.

We fully agree with these important comments of the Reviewer. Moreover, the suggested experiment has already been in the pipeline of our research program. Importantly, low levels of D₃ dopamine receptors have been reported in other brain regions, and cariprazine -as the Reviewer correctly pointed out and as we show in HEK 293 cells- can also bind to D₂ receptors albeit at substantially lower affinity. The Islands of Calleja is located ventrally to the nucleus accumbens/ventral striatum and to the dorsal striatum. While the former is known to express low levels of D₃ receptors and high levels of D₂ receptors, the latter contains only high density of D₂ receptors. To demonstrate that PharmacOSTORM is capable to detect low copy numbers of D₃ receptors and can also visualize other specific cariprazine binding sites that have lower binding affinity in a complex tissue preparation, we compared these three brain areas. In perfect agreement with the prior published data and our cell culture measurements, PharmacOSTORM imaging of the three adjacent brain areas revealed an order of magnitude difference in the binding density of fluo-cariprazine in the dorsal and ventral striatum compared to the Islands of Calleja. Moreover, the control experiments with D₃ knockout mice and unlabeled cariprazine pretreatment (displacement assay) could even quantitatively determine the magnitude of the differences in the density of D₃- and non-D₃ (presumably D₂) cariprazine binding sites. Again, we are very grateful to the Reviewer for suggesting this experiment because the obtained data

presented in new Supplementary Fig. S10 substantially improves both the methodical and biological aspects of the study. The importance of these control experiments for quantitative distinction of non-specific binding from functionally relevant molecular localizations that reflects true cariprazine target engagement sites is now discussed together with the rationale for optimizing washing steps for different ligands with distinct k_{off} values. Please note that the nearest neighbor data also depends on the abundance of the subcellular profiles carrying cariprazine binding sites that may affect the comparison between the different areas. Therefore, we performed density analysis.

- **Minor comment:**

The reader cannot estimate the degree of labelling in other brain regions based on the images in figure 3. Ideally, the authors would provide images without the DAPI merge (provides little information to the reader and impedes the reading of the images both a low and super resolution level) as well as images with dramatically increased contrast (saturating the signal in the Islands of Calleja) for panels 3a,b,f,g; 5b.

We thank the Reviewer for proposing these images that are presented now in Supplementary Figure S10a. Oversaturating the fluorescent signal in the Islands of Calleja permits the visualization of low levels of fluo-cariprazine binding in the adjacent areas, but excludes the quantitative measurement that could be performed with PharmacoS_TORM and shown in Supplementary Figure S10b-g. DAPI and fluo-cariprazine signals of multi-channel images (including those that were suggested by the Reviewer) are now shown separately in the raw data file.

REVIEWERS' COMMENTS

Reviewer #4 (Remarks to the Author):

In this manuscript revision the authors present a new set of data, discuss and elaborate on key technical steps of the PharmacoS_TORM approach. The density analysis in different striatum regions in WT and D3R KO alleviates my remaining concern regarding the signal to noise performance of PharmacoS_TORM in a very compelling fashion. The expended method procedure and discussion now ensure a proper use of the approach. I have no remaining concerns and can only support publication of the manuscript in its current form.

Reviewer #4:

In this manuscript revision the authors present a new set of data, discuss and elaborate on key technical steps of the PharmacoS_TORM approach. The density analysis in different striatum regions in WT and D3R KO alleviates my remaining concern regarding the signal to noise performance of PharmacoS_TORM in a very compelling fashion. The expended method procedure and discussion now ensure a proper use of the approach. I have no remaining concerns and can only support publication of the manuscript in its current form.

We are very grateful to the Reviewer for his/her supporting opinion.